

# Discrete differential geometry of fluvial landscapes

Nathaniel Klema[1,2], Leif Karlstrom[2], and Joshua Roering[2]

[1]Department of Physics and Engineering, Fort Lewis College, Durango, Colorado 81301, U.S.A.
[2]Department of Earth Sciences, University of Oregon, Eugene, Oregon 97403, U.S.A

**Correspondence:** Nathaniel Klema (ntklema@fortlewis.edu)

**Abstract.** Geomorphology as a discipline is often defined by the use of topographic geometry to understand surface processes on Earth and other planets. In practice this requires drawing quantitative connections between metrics of surface geometry and rates of exhumation, while also understanding the spatial partitioning of different erosion processes and the feedbacks between them. Many landscape evolution studies leverage curvature calculated as the scalar output of the Laplacian operator, which does not leverage all the information contained in the surface curvature tensor and which admits systematic error (up to $\sim 300\%$ percent) when applied directly to map-view topographic projections. In this study we use a formal surface theory approach to compute intrinsic and extrinsic curvature metrics, and associated shape-class distributions, of approximate steady-state fluvial topography of the Oregon Coast Range, USA. This workflow, including careful spectral filtering to isolate wavelengths of interest, provides a nuanced view of landscape structure, while simultaneously eliminating systematic errors arising from map-view approaches to topographic analysis. We leverage two invariants of the curvature tensor at a point – the Mean and Gaussian curvatures – to identify novel systematic structure of topographic geometry in channel and ridge networks that captures the full compliment of documented process regime transitions. Finally, we show remarkable symmetries in the distribution of Mean curvature and associated shape classes, specifically an equipartition of the landscape between concave-down and concave-up elements. These results suggest that formal surface theory approaches could prove valuable in maximizing the utility of digital elevation data and understanding the processes driving the evolution and organization of fluvial landscapes.

## 1 Introduction

The Earth's surface contains multi-scale signatures of the processes that have shaped it. Over length scales of $10^2 - 10^4$ km long-wavelength relief generally tracks patterns of lithospheric deformation and isostacy (Wieczorek, 2015) with relief generally increasing as the horizontal scale of measurement increases (Turcotte, 1987). The resulting gravitational gradients drive surface erosion, which shapes the landscape at finer scales (Perron et al., 2008; Hooshyar et al., 2020; Bonetti et al., 2020) through a combination of diffusive (Roering et al., 2001a), advective (Whipple and Tucker, 1999), and stochastic mass transport (Furbish et al., 2009). The rates of these erosion processes can be quantitatively linked to topographic form, and so geometry of the landscape can be used to study the competing long-timescale dynamics of bedrock uplift (Kirby and Whipple, 2012; Klema et al., 2023), and erosion that is modulated by lithology (Stock and Montgomery, 1999), climate (Ferrier et al., 2013), and ecology (Amundson et al., 2015).



In the spirit of reductionism, geomorphic studies often isolate regions where a single erosion process is assumed dominant, requiring the identification of distinct process regimes. There are many established approaches to landscape partitioning (Montgomery and Foufoula-Georgiou, 1993; Shary, 1995; Jasiewicz and Stepinski, 2013), however this approach comes at the risk of oversimplification. For example, the transition from hillslopes to fluvial channels often occurs just below topographic hollows
that where shallow landslides tend to initiate and the interactions between end-member process domains have implications for both landscape evolution and hazard prediction (Stock and Dietrich, 2003; Struble et al., 2023). As digital elevation models (DEMs) become increasingly high resolution in space and multi-temporal (Crosby et al., 2020), there is increasing opportunity to understand landscape development holistically through the development of quantitative tools that are accurate across all field settings and can resolve the details of such transitions.

However, common topographic analysis workflows use simplistic approaches to DEM analysis that do not extract the full richness of information contained in DEM data, and which introduce known systematic projection error that scales with both slope and curvature (Minár et al., 2020; Bergbauer and Pollard, 2003). The potential of differential geometry for improved DEM processing has already been established in several parallel earth science fields, having been used to calculate topographic stresses relevant to critical zone processes (Moon et al., 2017), model mechanisms of sheet joint development on bedrock surfaces
(Martel, 2011), and describe the structure of bedrock folds (Mynatt et al., 2007; Pearce et al., 2006). Recently, topographic contour curvature has been recognized as a key ingredient for gridscale-independent computation of flow accumulation and its role in landscape evolution models (Bonetti et al., 2018, 2020). However, widespread adoption of these techniques has been slow, perhaps because of a conceptual disconnect between resultant metrics of topographic geometry and area-space landscape partitioning frameworks that are at the core of landscape evolution theory.

With this in mind, here we develop a landscape classification workflow based on invariants of the curvature tensor that both removes known sources of systematic error in common curvature calculation methods (Minár et al., 2020) and provides a fully self-consistent means of calculating all common topographic metrics across process domains in discretely sampled DEMs. We apply our method to fluvial topography of the Oregon Coast Range, long taken to be a type setting for near-steady-state topography in which uplift is balanced by erosion everywhere. In addition to the practical benefits of accuracy and reproducibility
in computing curvature from DEMs, we demonstrate that this method provides insight into systematic geomorphic process transitions that shape landscapes from hilltops to channel networks.

## 1.1 Historical context

The connection between surface process rates and geometry was recognized as early as the late 19th century when work by G.K. Gilbert and W.M. Davis suggested connections between hillslope convexity and rates and styles of denudation in
mountain terrains (Gilbert, 1877; Davis, 1892). Subsequent work (Gilbert, 1908; Fenneman, 1908) showed that both slope and curvature near drainage divides varies systematically with erosion rate and noted that spatial partitioning of dominant erosion mechanisms results in regular geometric patterns toward which landscapes tend to evolve.



Efforts to define structures in topography predate these observations, however. As has been pointed out in Bonetti et al. (2018), topographic curvature as been studied since at least the middle nineteenth century: Arthur Cayley (Cayley, 1859) used topo-
graphic contours to show that watershed bounding ridges are composed of "summits" (we will term these structures "domes") connected by "knots" (we will call these "saddles") such that each ridge line contains one more "summit" than "knot". He argued that "immits" (we will call these "basins") would be similarly connected by bridging saddle structures such that there is one more "immit" than connecting saddle. James Clerk Maxwell (Maxwell, 1870) then argued that the Earth's surface could be classified as one of four shape classes; "hills" (domes), "dales" (basins), "passes" connecting hills (antiformal saddles),
and "bars" connecting dales (synformal saddles). Maxwell showed that continuity of the surface requires there to be one more dale than bar, and one more hill than pass, thus reaching the same conclusion as Cayley about the organization of topography without being restricted to ridges and channels.

We will show that a classification of topographic geometry as a function of upstream drainage area at a point, which is understood to reflect both deep landscape organization (Hack et al., 1957) and geomorphic process regimes (Flint, 1974; Montgomery
and Buffington, 1997; Kirby and Whipple, 2012), provides a quantitative connection between the early landscape organization theories of Maxwell and Cayley and area-space analysis methods common in fluvial geomorphology.

## 1.2 The use of curvature in geomorphology

Topographic curvature is used in geomorphology for surface classification and as an ingredient to mechanistic transport laws. For examples of classification, Shary (1995) derived 12 curvature metrics which were used in a landscape partitioning scheme,
Passalacqua et al. (2010) used geodesic curvature of topographic contours in combination with drainage area thresholding to extract channel networks from gridded Digital Elevation Models (DEMs), Bonetti et al. (2018) showed that curvature is intimately connected to the computation of upstream area, Minár et al. (2020) presented an extensive list of possible land surface curvature metrics and proposed possible links to topographic equilibrium, and Schmidt et al. (2003) derived curvature metrics using 2-d polynomial fits of topography for GIS applications. Such classification schemes have proven useful in surface
process studies (Sofia, 2020) and for mapping topographic characteristics of hazard susceptibility (Luu et al., 2024) and land use (Riza et al., 2022).

In mechanistic erosion models curvature arises from continuity requirements as the divergence of a gradient driven sediment flux law (Fernandes and Dietrich, 1997). Curvature is thus often used as a quantitative proxy for spatial variation in erosion rates (Struble et al., 2024). For example, at the scale of orogenic provinces, simple models of landscape relaxation in response
to uplift use versions of the heat equation where erosion rates are taken to be the product of long-wavelength surface curvature and an empirical diffusivity constant (Watts, 2001; Ruh, 2020). More detailed landscape evolution studies often leverage advection-diffusion equations where sediment transport within fluvial networks (Whipple and Tucker, 1999) is matched by curvature-driven diffusion of ridges and hillslopes (Roering et al., 1999; O'Hara et al., 2019). Adding a source term to represent bedrock uplift, a commonly used form of the landscape evolution equation arises as





$$\frac{dz}{dt} = U - KA^m|\nabla z|^n - D\nabla^2 z, \tag{1}$$

where $z$ is surface elevation, $U$ is uplift rate, $A$ is drainage area upstream of a given point (a proxy for stream discharge), and $K$, $m$, $n$, and $D$ are empirical constants that account for the geologic, hydrologic, and environmental factors that modulate rates of mass transport. The second term on the RHS is the 'stream power' model for bedrock fluvial erosion (Whipple and Tucker, 1999), while the third term models linear hillslope diffusion. Steady state refers to the situation where $dz/dt = 0$, in

which case equation 1 can be rewritten as

$$U = KA^m|\nabla z|^n + D\nabla^2 z. \tag{2}$$

Here uplift rate ($U$) is equal to the sum of the advection of material by the river network and hillslope diffusion, which dominates in unchannelized portions of the landscapes (hillslopes). If the coefficients in equation 2 ($U, K, m, n, D$; $A$ is always a nonlinear function of spatial position) do not vary spatially or temporally then variation in stream gradient and hilltop

curvature map variation in uplift (Wobus et al., 2006). Conversely, in regions where uplift is spatially uniform, variation in these coefficients can be used to infer the effects of climatic, lithologic, and environmental factors on the rate and style of erosion (Ferrier et al., 2013; Stock and Montgomery, 1999; Amundson et al., 2015).

An extensive literature has been established leveraging one or both of these limits (Perron et al., 2009; Whipple et al., 2013). However, the utility and reproducibility of this approach depends on accurate calculation from DEMs of curvature, slope, and

upstream catchment area.

## 2 Oregon Coast Range study site

We test our method of geometric classification in the central Coast Range, USA, a forearc landscape of the Cascades subduction zone. Our study area is a suite of $\sim 10$ km$^2$ basins that host fluvial and debris flow channel networks between the Umpqua and Smith River basins near Reedsport, Oregon. Bedrock in this study area is composed entirely of the Tyee Formation (Baldwin,

1961; Beaulieu and Hughes, 1975), a 3 km thick suite of accreted Eocene turbidites that was subject to uplift during the Miocene (McNeill et al., 2000; Wells et al., 2014) and continues to be uplifted today with long-term rates ranging from $0.05$ mm yr$^{-1}$ to over $0.4$ mm yr$^{-1}$ (Kelsey et al., 1996; Personius, 1995).

The Coast Range has long been identified and studied as an archetypal steady-state landscape due to its uniform ridge-valley topography (Dietrich and Dunne, 1978; Montgomery, 2001), and documented correlations between catchment averaged erosion

rates, uplift rates, and topographic proxies for erosion rate (Reneau and Dietrich, 1991; Heimsath et al., 2001; Struble et al., 2024). We focus on a small portion of the Coast Range where we do not expect to see the variation in lithology (Baldwin, 1961;



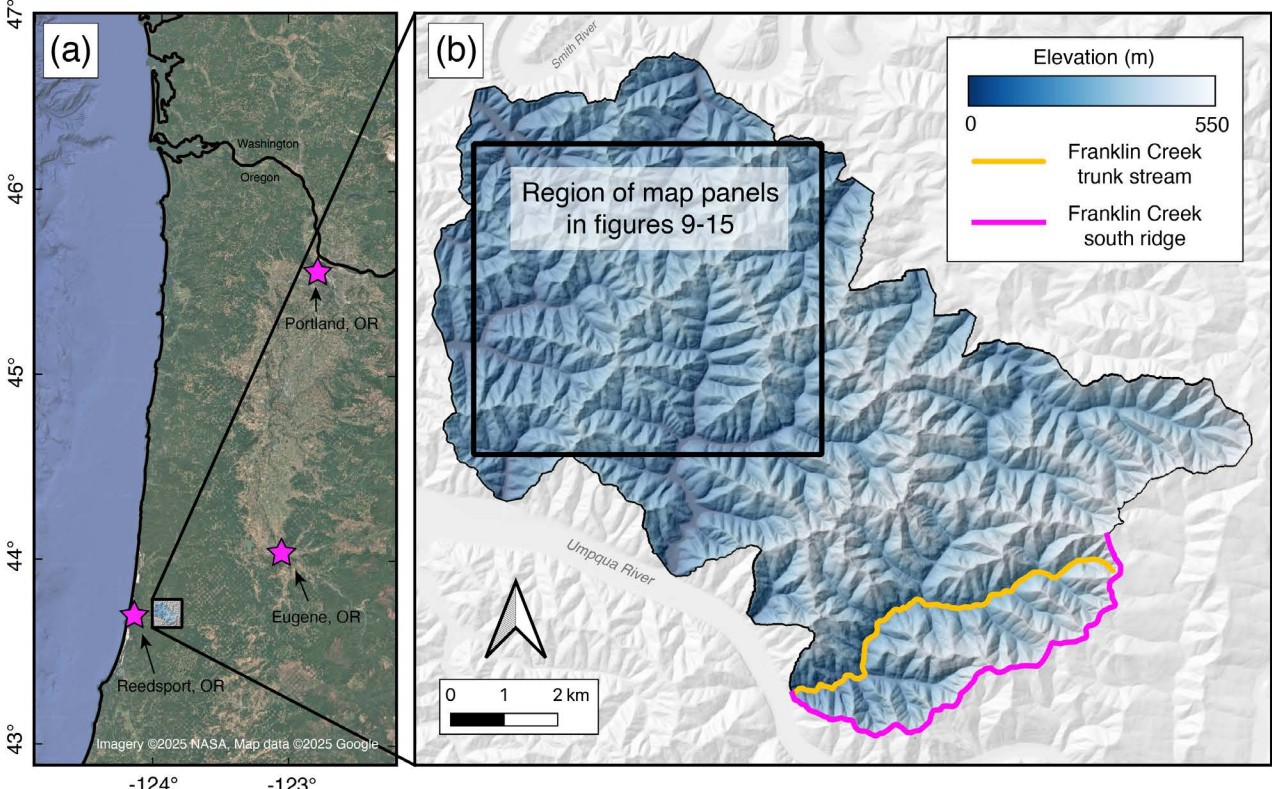

**Figure 1. Map of study area. a.** Overview map of Cascadia coastal region showing location of study site. Satellite imagery from Google Earth, accessed through QGIS XYZ tiles on June 13th, 2025. **b.** Elevation map of study area showing location of the Franklin Creek trunk stream and southern ridge of Franklin Creek basin analyzed in section 7.3. Black outline shows region of focused maps in figures 9-15

Beaulieu and Hughes, 1975) or climate (Daly and Bryant, 2013). Owing to the relatively gentle dip of the bedrock, this area is not subject to deep-seated landslides that interrupt characteristic ridge-valley terrain in other portions of the Coast Range (Roering et al., 2005; LaHusen et al., 2020).

## 3   Intrinsic versus extrinsic topographic curvature

The term "curvature" formally refers to a class of mathematical operations that describe the degree to which a surface (or more generally, a manifold) deviates from being planar (Needham, 2021). The tools of differential geometry and tensor calculus were in part developed to describe these operations. Measures of curvature can be classified as either 'intrinsic' or 'extrinsic' where intrinsic curvatures are invariant qualities of a surface, meaning their value is independent of external coordinate systems and



can be calculated using only local surface information (Needham, 2021). Conversely, extrinsic measures are built using external coordinates and so their values depend on orientation of a surface within a given reference frame (Struik, 1950; O'Neill, 2006; Minár et al., 2020).

Many topographic analysis workflows calculate curvature as an extrinsic scalar quantity given by the Laplacian $\nabla^2 z(x, y)$ where $z$ is elevation, $\nabla$ is the gradient operator, and the reference frame is given by east-west coordinates of DEM pixels.

Usually this quantity is either calculated directly from gridded DEMs or polynomial fits of elevation data (Hurst et al., 2012; Schmidt et al., 2003; Roering et al., 1999; Perron et al., 2009). While theoretically the Laplacian gives a curvature value proportional to the intrinsically defined Mean Curvature (section 5.3), these approaches admit systematic slope-dependent error (Bergbauer and Pollard, 2003; Minár et al., 2020) in addition to error that stems from the projection of topography onto a regularly sampled 2-d grid. Instead a DEM is must be viewed as a set of irregularly spaced data points sampling a 2-d surface

embedded in a 3-dimensional space. This distinction is important because, while map-view projections of DEMs assume uniform distances and angles between grid points, lines on a topographic surface change length and orientation with changing curvature and slope resulting in a breakdown of euclidean assumptions on which the Laplacian method is based (Needham, 2021).

The effects of the projection process can be seen in Fig. 2, which compares the distances and angles of a map projection (Fig. 140 2.a) to those of the same grid lines overlain on a 3-d representation of the surface (Fig. 2.a). We see that the E-W and N-S grid lines in Fig. 2.a) are not perpendicular on the surface Fig. 2.b) and change orientation between grid nodes. As a result grid cells do not have uniform dimensions and can not be described with a universal coordinate system. One way to visualize this is to imagine walking from point $p$ along displacement vectors defined by the map-view grid lines. In the map view projection it is clear that one would arrive at point $\hat{q}$ located a distance $d\hat{s}$ from the origin, however in fact one would travel the distance

$ds$ arriving at point $q$. The effects of this distortion on the accuracy of curvature, slope, and drainage area calculations will be explored quantitatively in section 6.

The problem of projection error for topographic maps was recognized by Leonard Euler in 1775 and motivated the work of Carl Frederick Gauss who, roughly fifty years after Euler's observation, established the modern method for calculating the curvature of surfaces (Gauss, 1902; Needham, 2021). Gauss recognized that, while Euclidean spaces exhibit global parallelism (parallel 150 lines remain parallel for ever; Fig. 2.a) curved surfaces do not. Gauss addressed this by defining a locale, non-orthogonal, coordinate system at each point (the dashed lines in Fig. 2.b for point p) and discovered he could characterize local variations in the surface via the coefficients of two quadratic equations known as the first and second fundamental forms. This discovery paved the way for the fundamental theorem of surface theory, proved in 1867 by Pierre Ossian Bonnet (Cogliati and Rivis, 2022). This theorem states that all qualities of a surface, with the exception of its orientation in space, can be derived using the

6 coefficients of the first and second fundamental forms.

During the same time period Bernhard Riemann, a student of Gauss, set the stage for modern differential geometry by generalizing Gauss' ideas to higher dimensional spaces and showing that characterizing curvature of an $n$-dimensional manifold





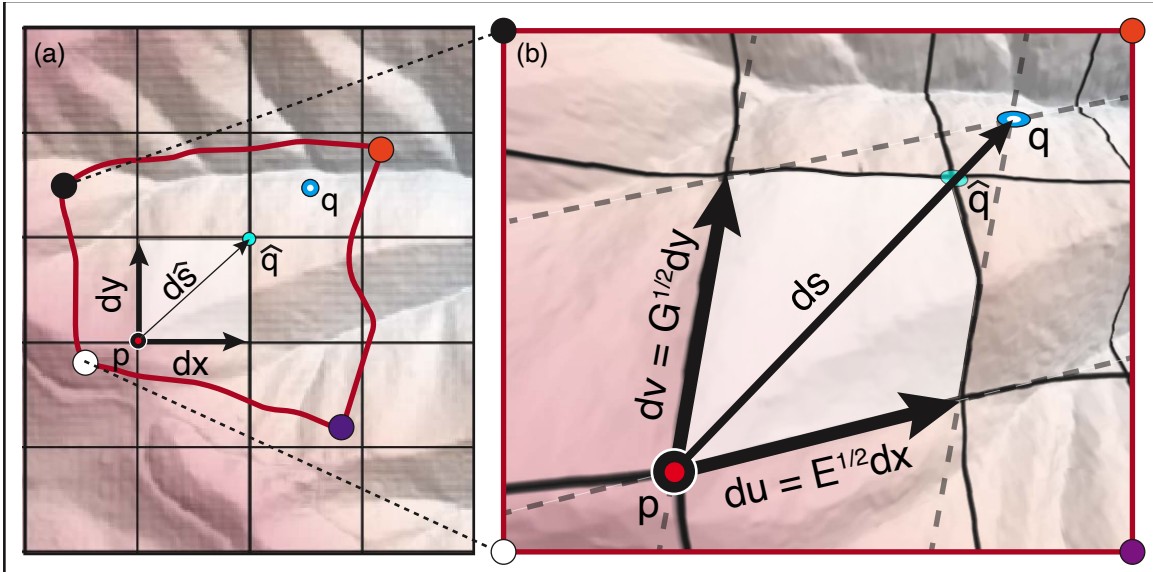

**Figure 2. Difference between distances and angles measured on a map projection versus on the surface. a.** Map projection of DEM including map grid defined by E-W and N-S lines with grid spacing $dx$. The red line corresponds to the rectangular outline of panel b. **b.** DEM viewed as a 2-d manifold embedded in a 3-d space. Dashed lines how a locally defined $u$-$v$ coordinate system that follows $x$ and $y$ curves on the map projection, but which are not orthogonal or of equal length due to surface distortion. $E$ and $G$ are coefficients of the first fundamental form, and $ds$ is the displacement vector that results moving one grid space along each of these coordinate vectors

requires $\frac{1}{12}n^2(n^2-1)$ distinct coefficients (Pesic, 2007). Thus while a single measure of curvature (e.g. the Laplacian) is sufficient to characterize a 2-d map projection of topography, accurate representation of 3-d topography requires six distinct components; the coefficients of the first and second fundamental forms. It is worth noting that most modern treatments of curvature, including profound discoveries in physics of the 20th century (e.g. general relativity; Einstein and Lawson (2013)), use the tools of tensor calculus for which these six coefficients are the independent components of the 'Riemannian curvature tensor' Needham (2021). However, the original theory is sufficient for our analysis of topography, so we use Gauss' classical definition of surface curvature here.

## 4 Spectral filtering of gridded datasets

In order to calculate the curvatures of DEMs it is necessary to do some degree of smoothing to both remove artifacts of the gridding process (Reuter et al., 2009; Bui and Glennie, 2023; Bater and Coops, 2009) and generate a mathematically continuous surface that satisfies the differentiability requirement of calculus (Stewart, 2003). For this study we select $8.1$ m resolution DEM data freely available through the National Map (https://apps.nationalmap.gov/downloader/). While higher resolution LiDAR (Light Detection and Ranging) data are available in the study area we select the courser dataset, which gives substantially shorter computation times while still resolving geometric trends at the scale of fluvial basins.



Numerous tools are available for smoothing digital topographic data, including b-spline fitting (Brigham and Crider, 2022), wavelets (Struble et al., 2024), selective denoising (Gallant, 2011), and TIN inerpolation (Jordan, 2007). We choose to filter the data using a Discrete Fourier Transform (DFT; also a contribution of Gauss (Heideman et al., 1985)), which decomposes

discretely sampled signals into sums of harmonic functions. Smoothing is accomplished via low-pass filter, where information at wavelengths smaller than a defined cutoff is removed. Fourier methods have been extensively applied in geomorphology including the identification of characteristic process scales (Perron et al., 2008), landform identification (Booth et al., 2009), and to assess topographic controls on mass transport mechanics (Richardson and Karlstrom, 2019; Black et al., 2017; Crozier et al., 2018).

One challenge of Fourier methods in topographic analysis is that harmonic functions do not naturally respect the finite nature of a DEM in which boundaries are non-homogeneous. Tapering of the data is thus required to obtain zero elevation at the boundaries prior to DFT. It is common to accomplish this by convolving the DEM grid with a 2-d raised cosine (aka Hanning window), such that the resulting topography is equal to its actual value only in center of the grid, and is elsewhere subdued by an amount that scales with distance towards the margins (Perron et al., 2008). A downside of this approach is that it diminishes the

spectral power of relevant landscape features, and introduces artificial long-wavelength signals that are difficult to differentiate from real structures in the resulting power spectrum.

Fortunately, solutions can be found (Harris, 1978; Kirby, 2014) . For example Mcnutt (1983) showed that spectral contamination can be minimized by first reflecting the topographic grid along each coordinate axis, then tapering the data only in the reflected portions that fall outside the limits of the original DEM. This avoids the introduction of distortion within the study

area through the windowing process, and only introduces artificial signals with significant power that are at or above the scale of the full DEM extent.

We adopt this mirroring approach and apply a Tukey window, which consists of a boxcar function convolved with a cosine taper along the margins (Harris, 1978). We apply the default Tukey window in the *window2* function in Matlab and find that this method minimizes the introduction of spurious signals in the pre-processing step.

The Discrete Fourier Transform (DFT) is calculated as

$$Z(k_x, k_y) = \sum_{p=0}^{N_x-1} \sum_{q=0}^{N_y-1} z(p\Delta x, q\Delta y)e^{-2\pi i(\frac{k_x p}{N_x} + \frac{k_y q}{N_y})} \tag{3}$$

where $N_x$ and $N_y$ are the number of grid cells in each direction, $p$ and $q$ are array indices, $\Delta x$ and $\Delta y$ are the grid spacings in each direction, and $k_x$ and $k_y$ are the wavenumbers in the respective x and y directions (Perron et al., 2008). Each value in the output array given by the above equation is associated with a frequency in $x$ and $y$ directions with magnitudes

$$f_x = \frac{k_x}{N_x \Delta x}, \qquad f_y = \frac{k_y}{N_y \Delta y}. \tag{4}$$

These frequencies can then be used to define a radial frequency as

$$f_r = \sqrt{(f_x^2 + f_y^2)}. \tag{5}$$



The DFT periodogram is then given by

$$P_f(k_x, k_y) = \frac{1}{N_x^2 N_y^2} |Z(k_x, k_y)|^2 \tag{6}$$

Following Perron et al. (2008) we design a half-Gaussian filter based on radial frequencies defined as

$$F_{low} = \begin{cases} 1, & f < f_1 \\ \exp\left(\frac{-(f-f_1)^2}{2\sigma^2}\right), & f \geq f_1 \end{cases} \tag{7}$$

where $\sigma = \frac{1}{3}|f_2 - f_1|$ is the standard deviation. The filter is convolved with the radial frequency spectrum to suppress frequencies outside the filter tapering window. The filtered spectrum is then reversed transformed, and the original domain of the DEM is extracted from the windowed representation to yield a low-pass filtered raster of topography.

## 5  Calculation of curvature

In this section we derive the coefficients of Gauss' two fundamental forms, and use them to calculate curvature metrics used in our topographic analysis. While there are several methods for deriving these quantities, we base our derivation on that of Struik (1950) (Chapter 2). See also Bergbauer and Pollard (2003), Mynatt et al. (2007), and Martel (2011) for similar geologically motivated derivations. Note that we use prime notation for partial derivatives relative to directions given by subscript indices
$(\mathbf{r}'_u = \partial \mathbf{r}/\partial u)$.

### 5.1  Deriving the first and second fundamental forms

Surface curvature at a given point can be defined by considering an arbitrary path drawn over the surface. Infinitesimal displacements along such a path trace out arcs that lie within a plane containing the unit tangent ($\mathbf{t}$) and normal ($\mathbf{N}$) vectors of the curve (known as the 'osculating plane'). We define a position vector to a point on the surface in 3-d cartesian space as

$$\mathbf{r} = \mathbf{r}(u,v) = r_1\mathbf{e_1} + r_2\mathbf{e_2} + r_3\mathbf{e_3}, \tag{8}$$

where the parameters $u$ and $v$ represent two intersecting curves on the surface that define a local coordinate system, and $\mathbf{e_1}$, $\mathbf{e_2}$, and $\mathbf{e_3}$ are Cartesian basis vectors. The choice of $u$ and $v$ is completely arbitrary except for the requirement

$$\mathbf{r}'_u \times \mathbf{r}'_v \neq 0 \tag{9}$$





which ensures that a normal vector can be defined. In our case the the $u$ and $v$ curves follow the E-W and N-S DEM grid lines.

For any small displacement ($ds$) a tangent vector ($\mathbf{t}$) and unit normal vector ($\mathbf{N}$) can be defined in terms of the $u, v$ curves as

$$\mathbf{t} = d\mathbf{r}/ds \tag{10}$$

and

$$\mathbf{N} = \frac{\mathbf{r}'_u \times \mathbf{r}'_v}{|\mathbf{r}'_u \times \mathbf{r}'_v|}. \tag{11}$$

respectively, where $d\mathbf{r} = \mathbf{r}'_u du + \mathbf{r}'_v dv$ is the change in position resultant of a small displacement on the surface. Equation 11

is the unit normal of the tangent plane ($\mathbf{N} = N_1\mathbf{e_1} + N_2\mathbf{e_2} + N_3\mathbf{e_3}$) with the slope of the tangent plane given by

$$S_T = \left| \tan\left( \frac{\pi}{2} - \sin^{-1}(N_3) \right) \right|. \tag{12}$$

The magnitude of a displacement along the curve is given by the Pythagorean Theorem as

$$I = ds^2 = d\mathbf{r} \cdot d\mathbf{r} = E du^2 + 2F dudv + G dv^2. \tag{13}$$

Equation 13 is Gauss' first fundamental form ($I$), also called the surface metric formula. It is a measure of distances on

the surface, where $E = \mathbf{r}'_u \cdot \mathbf{r}'_u$, $F = \mathbf{r}'_u \cdot \mathbf{r}'_v$, and $G = \mathbf{r}'_v \cdot \mathbf{r}'_v$ (the metric coefficients) quantify the proportionality of distances measured on the surface to distances in the projected cartesian reference frame (Needham, 2021). The metric coefficients also allow us to calculate the ratio of surface area on topography to the area of a grid cell via the relation

$$\alpha = \sqrt{EG - F^2}. \tag{14}$$

We will use this 'area expansion factor' ($\alpha$) to compare intrinsic versus extrinsically derived drainage area in section 6. Along

any surface curve $ds$ that locally follows the shortest path between points (geodesic) the only curvature will be the 'normal curvature' ($\kappa$) which measures rotation of the surface tangent along the curve. Mathematically this can be defined as the projection of the full derivative of the unit tangent onto the unit normal written

$$\frac{d\mathbf{t}}{ds} = \left( \frac{d\mathbf{t}}{ds} \cdot \mathbf{N} \right)\mathbf{N} = \kappa\mathbf{N}. \tag{15}$$

It follows that the normal curvature is

$$\kappa = \frac{d\mathbf{t}}{ds} \cdot \mathbf{N} = -\mathbf{t} \cdot \frac{d\mathbf{N}}{ds} \tag{16}$$

where $d\mathbf{N} = \mathbf{N}'_u du + \mathbf{N}'_v dv$. Plugging in equation 10 as the definition of the tangent vector, equation 16 can be rewritten as

$$\kappa = -\frac{d\mathbf{r}}{ds} \cdot \frac{d\mathbf{N}}{ds} = -\frac{e du^2 + 2f dudv + g dv^2}{E du^2 + 2F dudv + G dv^2} = -\frac{II}{I}. \tag{17}$$

The expression in the numerator

$$II = d\mathbf{r} \cdot d\mathbf{N} = e du^2 + 2f dudv + g dv^2 \tag{18}$$





is the second fundamental form ($II$) with coefficients $e = \mathbf{r}''_{uu} \cdot \mathbf{N}$, $f = \mathbf{r}''_{uv} \cdot \mathbf{N}$, and $g = \mathbf{r}''_{vv} \cdot \mathbf{N}$ that measure changes in the orientation of the tangent plane along $ds$.

Equation 17 allows us to completely define the shape of a complicated surface in 3-d space using a cartesian coordinate system, but without the assumption of orthogonality on the surface that leads to systematic error in the Laplacian. The coefficients of the second fundamental form ($e, f$ and $g$; Equation 18) are the directional curvatures where $e$ and $g$ correspond to curvature along the E-W and N-S grid line respectively, and $f$ is a cross term that accounts for the lack of orthogonality of these directions on the surface. These curvatures are scaled by the coefficients of the first fundamental form ($E, F$, and $G$; Equation 13), which maps lengths on the coordinate grid to lengths on the surface accounting for the effect of slope on surface distances and areas. Together these six coefficients allow for the rigorous classification of topographic surfaces that meets criteria set forth by the fundamental theorem of surface theory, and allows us to compute topographic curvatures in a way that minimizes known distortion inherent in extrinsic surface classification methods.

## 5.2 Finding the principal curvatures

Equation 17 is a general expression for curvature for a single path over the surface. The existence of infinite paths through any given point means there are infinite measures of curvature possible, however there always exist two perpendicular paths along which the maximum and minimum curvatures are found. This observation is attributed to Euler, who showed that as the osculating plane rotates about an axis defined by the surface normal vector $\mathbf{N}$ of a continuous surface the normal curvature ($\kappa$; equation 17) varies as

$$\kappa(\theta) = k_1 \cos^2 \theta + k_2 \sin^2 \theta, \tag{19}$$

where $k_1$ and $k_2$ are the extrema, termed the two 'principal curvatures', and $\theta$ is angular direction measured within the surface tangent plane. Equation 19, known as Euler's Theorem, shows that once the principal curvatures are known they can then be used to calculate curvature along any path over the surface (not to be confused with the 'Eulers Theorem' in number theory or the 'Euler Identity' of complex analysis). Fig. 3 shows a plot of Equation 19 noting the location of principal curvatures and the Mean Curvature, which will be discussed further in the following section. To find the directions of the two principal curvatures we first define a new parameter $\lambda = dv/du$, allowing us to rewrite Equation 17 in terms of a single variable as

$$\kappa = \frac{edu^2 + 2f\lambda du^2 + g\lambda^2 du^2}{Edu^2 + 2F\lambda du^2 + G\lambda^2 du^2} = \frac{e + 2f\lambda + g\lambda^2}{E + 2F\lambda + G\lambda^2}. \tag{20}$$

The principal curvatures correspond to extrema where $d\kappa/d\lambda = 0$, so we differentiate Equation 20 with respect to $\lambda$ and set the result equal to zero giving

$$\frac{d\kappa}{d\lambda} = (E + 2F\lambda + G\lambda^2)(f + g\lambda) - (e + 2f\lambda + g\lambda^2)(F + G\lambda) = 0. \tag{21}$$

Rearranging Equation 21 gives

$$(Fg - Gf)\lambda^2 + (Eg - Ge)\lambda + (Ef - Fe) = 0, \tag{22}$$





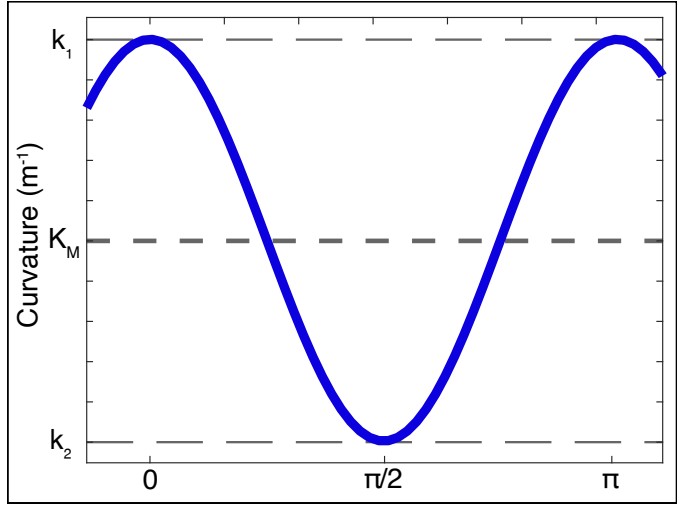

**Figure 3. Curvature as a function of the angle between the osculating plane and a reference direction.** Here the angle $\theta = 0$ is associated with the first principal curvature ($k_1$). $k_2$ is the second principal curvature and $K_M$ is the Mean curvature about which the value oscillates.

a quadratic equation in $\lambda$ with roots that correspond to the principal curvature directions. Recalling that $\lambda = dv/du$ we can then convert these principal directions into angles defined in our local $u$-$v$ coordinate system.

The magnitudes of the principal curvatures can similarly be found by finding a quadratic equation in the parameter of interest, in this case the normal curvature $\kappa$. Given the condition $d\kappa/d\lambda = 0$ at curvature extrema equations 20 and 21 can be combined to give a simpler expression for the curvature

$$\kappa = \frac{f + g\lambda}{F + G\lambda}. \tag{23}$$

Recognizing that

$$E + 2F\lambda + G\lambda^2 = (E + F\lambda) + \lambda(F + G\lambda) \tag{24}$$

and

$$e + 2f\lambda + g\lambda^2 = (e + f\lambda) + \lambda(f + g\lambda) \tag{25}$$

we show that

$$\frac{f + g\lambda}{F + G\lambda} = \frac{e + f\lambda}{E + F\lambda} \tag{26}$$

and so we arrive at two equivalent expressions for defining the curvature

$$\kappa = \frac{f + g\lambda}{F + G\lambda} = \frac{e + f\lambda}{E + F\lambda}. \tag{27}$$





The two expression for curvature given by Equation 27 can be rearranged as

$$f - \kappa F + \lambda(g - \kappa G) = 0 \tag{28}$$

and

$$e - \kappa E + f - \kappa F = 0 \tag{29}$$

respectively. If we then multiply both equations through by $du$ (remembering that $\lambda = dv/du$) then we arrive at a system of equations in terms of our original $u$-$v$ coordinate system

$$\begin{bmatrix} e - \kappa E & f - \kappa F \\ f - \kappa F & g - \kappa G \end{bmatrix} \begin{bmatrix} du \\ dv \end{bmatrix} = \begin{bmatrix} 0 \\ 0 \end{bmatrix}. \tag{30}$$

This has a non-trivial solution if the determinant of the coefficient matrix is zero, which corresponds to

$$(EG - F^2)\kappa^2 - (gE - 2fF + eG)\kappa + (eg - f^2) = 0, \tag{31}$$

a quadratic equation in $\kappa$. The roots of Equation 31 are the principal curvatures of the surface. The more positive principal curvature is typically called $k_1$, while the less positive curvature is $k_2$. In this work positive curvatures are defined as concave down.

It should be noted that in more modern literature the information contained in the fundamental forms is often presented in the Shape Operator, which is the second order tensor given by taking the negative gradient of a normal vector along an arbitrary tangent. This is written mathematically as

$$S(\mathbf{t}) = -\nabla_{\mathbf{t}} \mathbf{N} \tag{32}$$

where $\nabla_{\mathbf{t}}$ is the gradient defined along tangent vector $\mathbf{t}$ (O'Neill, 2006). In terms of our $u$-$v$ coordinate curves the first and second fundamental forms can be defined via this approach as $I(\mathbf{u}, \mathbf{v}) = \mathbf{u} \cdot \mathbf{v}$ and $II(\mathbf{u}, \mathbf{v}) = S(\mathbf{u}) \cdot \mathbf{v}$.

The information contained in the first fundamental form is used to construct the 'metric tensor'

$$I = \begin{bmatrix} E & F \\ F & G \end{bmatrix}, \tag{33}$$

while the information of the second fundamental form stored in the 'surface curvature tensor'

$$II = \begin{bmatrix} e & f \\ f & g \end{bmatrix}. \tag{34}$$

The Shape Operator is then given by

$$S = I^{-1} II. \tag{35}$$

The eigenvalues of the Shape Operator are the principle curvatures and the eigenvectors define the directions associated with these curvatures. The Gaussian and Mean curvatures, which we will define in the following section, are the determinate and trace of this matrix respectively.





### 5.3 Mean and Gaussian Curvatures, General Surface Shape Class categories

Once the principal curvatures are found, they can be used to calculate two other useful curvature metrics; the 'Mean' and 'Gaussian' curvatures. The Mean curvature follows directly from Euler's Theorem and is the value about which the curvature oscillates as a function of angle on the surface (Equation 19; Fig. 3).

We calculate the Mean curvature ($K_M$) as the average of the principal curvatures

$$K_M = \frac{k_1 + k_2}{2}, \tag{36}$$

although it actually does not require any knowledge of the principal curvatures directly and can be calculated as the average curvature of any two orthogonal paths through a given point, an idea we will revisit in section 6 on error in Laplacian curvature. Mean curvature can also be calculated as the trace of the Shape Operator tensor.

The Gaussian curvature measures an intrinsic quality of surfaces that is invariant under spatial transformations that conserve distance (isometries). This means its value does not depend on the shape of the surface, but captures a more subtle quality, which is the degree of stretching or bending required to deform a flat plane so that it conforms to a given surface. We compute the gaussian curvature as the product of the principal curvatures

$$K_G = k_1 k_2, \tag{37}$$

though more fundamentally it is the determinate of the Shape Operator tensor.

We note that while we will consistently refer to this quantity as the Gaussian curvature ($K_G$), in the literature it is also called the intrinsic curvature, total curvature, or sometimes just 'the curvature' in surface theory contexts (Needham, 2021).
The Mean and Gaussian curvature together can be used to classify the geometry about a point into the 8 distinct shape classes shown in Fig. 4. Since the Gaussian curvature is the product of the two principal curvatures it will only be positive in instances

where $k_1$ and $k_2$ have the same sign. This tells us that when $K_G$ is positive a surface is either a dome or basin, though because the Gaussian curvature is isometrically invariant it does not contain information about surface orientation. If $K_G$ is negative then $k_1$ and $k_2$ necessarily have opposing signs and so the surface is locally a saddle, though again with an orientation in space that cannot be uniquely determined by the Gaussian curvature.

If either $k_1$ or $k_2$ is equal to zero then $K_G$ is also zero. This case itself comprises its own class of surfaces, known as developable

surfaces, which can be formed from a plane without altering surface area and thus are intrinsically flat. 'Relatively developable surfaces' can be extracted from the landscape by assigning a zero value to curvatures under a defined threshold (Mynatt et al., 2007). Curvature thresholding to extract developable forms is a promising approach for classifying the distribution of landforms on the Earth's surface based on curvature. However, we do not explore this further here.

The orientation of a shape in space can be determined from the Mean curvature, and so allows us to put $K_G$ based classifications

into a landscape reference frame. $K_M$ is positive in two cases; when either both $k_1$ and $k_2$ are positive, or the higher magnitude curvature ($k_1$) is positive. This means that points in the landscape with $K_M > 0$ are concave down and are locally either domes





**Figure 4. Shape classes into which points on the surface can be sorted based on the signs on the Mean ($K_M$) and Gaussian ($K_G$) curvatures**. In this analysis we focus on those classes that can be assigned based on raw curvature values, which are synformal saddles, antiformal saddles, basins, and domes, and do not include developable surfaces or perfect saddles. Modified from Mynatt et al. (2007)

or antiformal saddles. Similarly if $K_M$ is negative then the orientation of the surface must be dominantly concave up, and so the surface is either a basin or synformal saddle.

In cases where $k_1$ and $k_2$ are equal and opposite the surface is a perfect saddle. This too comprises its own special class of surfaces, known as 'minimal surfaces', which have been largely tied to the optimization of surface area in self-organizing systems such as soap foams, cell membranes, and crystal lattices (Osserman, 2013). The existence of approximate perfect saddles could be explored in a geomorphologic context by assigning a threshold below which curvature variation is assumed negligible. Again, we do not endeavor to explore this here.

Instead, we focus on the 4 basic shape classes that can be assigned based on raw curvature calculations. This will provide us with sufficient information to quantify patterns of curvature in landscapes and relate these to geomorphic process regimes.

## 5.4 Computing curvatures on gridded DEMs

An approach to surface curvature calculation for gridded DEM data that utilizes the shape operator can be found in Mynatt et al. (2007) and Pearce et al. (2006), who apply a similar approach to study structural geologic surfaces. While the mathematical simplicity of this approach is appealing, it is computationally expensive to calculate and the Shape Operator and take the trace and determinate. This in practice limits the scale over which the methods can be applied. For example, on a personal laptop the calculation of curvatures on a $1000 \times 1000$ cell DEM using a MATLAB code similar to that employed by Mynatt et al. (2007) took $\sim 100$ times longer than our approach. Given the size of topographic datasets, decreasing computation times makes the tool much more practical for landscape scale analysis.





Once the landscape has been filtered all of the curvature metrics outlined in sections 5.2 and 5.3 are calculated for every
pixel in the DEM and binned by upstream drainage area. Area is calculated at each point in the DEM using the D-infinity
algorithm (Tarboton, 1997) implemented in the TopoToolbox MATLAB function library (Schwanghart and Scherler, 2014).
As our approach provides a way to quantify surface area associated with each DEM via the area expansion factor $\alpha$ (Equation
14) we weight DEM pixels such that we calculate the actual upstream surface area of topography rather than the area of the
map-view projection. We will quantify the difference between this and the standard flow routing approach in section 6.3.

However, before exploring the landscape curvature distribution we first must choose an appropriate scale of filtering. This is
accomplished by smoothing the landscape to progressively longer wavelengths, and assessing how our measures of curvature
vary with scale. To eliminate any high wavenumber gridding artifacts we first smooth the landscape to 50 m, then increase
the low-pass filter cutoff by increments of 50 m up to 500 m. The results can be seen in Fig. 5, which shows the land surface
gradient and the Gaussian and Mean curvatures binned by drainage area. While curvature and slope magnitudes vary with
increasing low-pass filter cutoffs, general trends in these metrics are similar across this range of filter cutoffs, making the
selection of a particular scale for our analysis somewhat arbitrary. However, we note that at cutoffs greater than $\sim 200$ m peaks
in the slope curves start to get pulled to longer wavelengths suggesting a shift in the structures being resolved at those scales.
We also note that, while the magnitude of Mean curvatures decreases systematically with increasing filter cutoff the the main
extrema in the Gaussian curvature have highest magnitudes at a cutoff of 200 m, perhaps indicating a characteristic curvature
scale in the landscape.

We perform all further analysis on topography low-pass filtered to 200 m, acknowledging that this choice is somewhat non-
unique. This filter scale allows us to identify landscape features that span hillslope and fluvial process regimes, however inhibits
our ability to see the effects of finer scale processes (e.g., tree throw; Roering et al. (2010)). Map-view distributions of Mean
and Gaussian curvatures, principal curvatures, tangent plane slope, and upstream drainage area are shown in Fig. 6.

## 6    Comparison with common methods of topographic geometry calculation

Applying Equations 1 and 2 to real topography requires calculation of curvature ($\nabla^2 z(x,y)$), slope ($\nabla z(x,y)$), and upstream
drainage area ($A(x,y)$) from elevation data. In this work we have proposed a self-consistent approach for computing all of
these metrics. We will now compare the outputs of our methods with approaches common in the geomorphology literature in
an effort to identify sources of systematic error.

### 6.1    Systematic error in Laplacian curvature

As outlined in section 3 geomorphologists often take 'curvature' to be synonymous with the output of the Laplacian operator
$\nabla^2 z(x,y) = z''_{xx} + z''_{yy}$, where $x$ and $y$ are horizontal coordinates (generally easting and northing) and $z$ is elevation. We can
understand the relationship between the Laplacian and $K_M$ through consideration of Euler's Theorem (Equation 19;l Fig. 3),
which shows that $K_M$ is the average of curvatures measured along any two orthogonal paths. Thus, while our approach uses





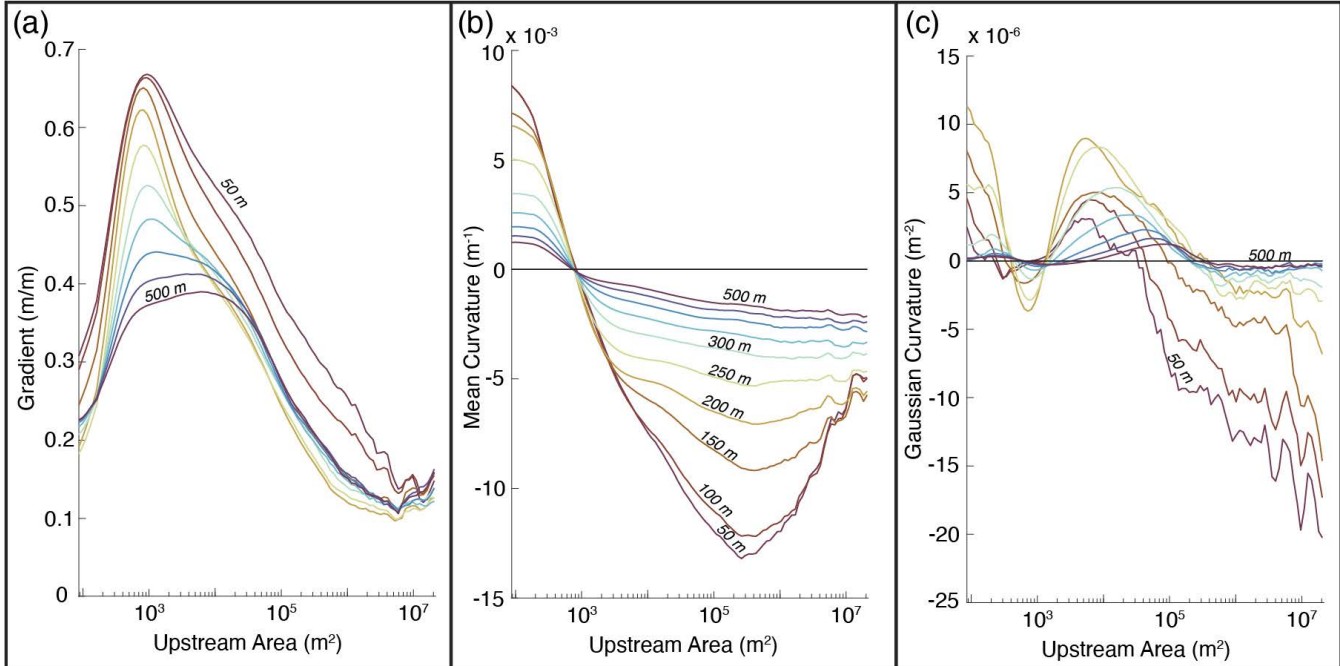

**Figure 5. Surface geometry metrics binned by upstream drainage area for a range of low-pass filter cut-offs between 50 m and 500 m calculated on 50 m intervals. a.** Gradient of tangent plane. **b.** Mean curvature ($K_M$). **c.** Gaussian curvature ($K_M$).

k₁ and k₂, knowledge of the principal curvatures is not required to calculate $K_M$. This means, if the $x$ and $y$ coordinate vectors are perpendicular on the surface, the relationship between the Laplacian and Mean curvatures is simply

$$K_M = \frac{z''_{xx} + z''_{yy}}{2} = \frac{1}{2}\nabla^2 z. \tag{38}$$

Leveraging Equation 38 allows for quantification of systematic error in the Laplacian method, which comes both from the effect of slope on derivative calculations (Bergbauer and Pollard (2003) and references therein) and the non-orthogonality of coordinate vectors on the topographic surface (section 3; Fig. 2).

We evaluate the significance of these errors by calculating Laplacian and Mean curvatures on a unit hemisphere generated on a $1001 \times 1001$ grid in MATLAB. Since this synthetic surface has a known Mean curvature ($K_M = 1/R = 1\ m^{-1}$) we can compare outputs of both the $1/2$ Laplacian and Mean curvature calculated on the grid to the analytic prediction to evaluate the accuracy of both methods. The Laplacian is calculated using the MATLAB 'del2' function, while the Mean curvature is derived using the methods of this paper. The results of this exercise can be found in Fig. 7. Fig. 7.b compares computed curvatures along the plane curve at the intersection between the sphere and the $y - z$ plane. We see that while $K_M$ calculated using our approach (blue dashed line) maintains accuracy along the curve, the 1/2 Laplacian (solid black line) converges to the actual value only at the center of the sphere, and that error (red dashed line) increases outward with increasing slope (purple dashed curve).





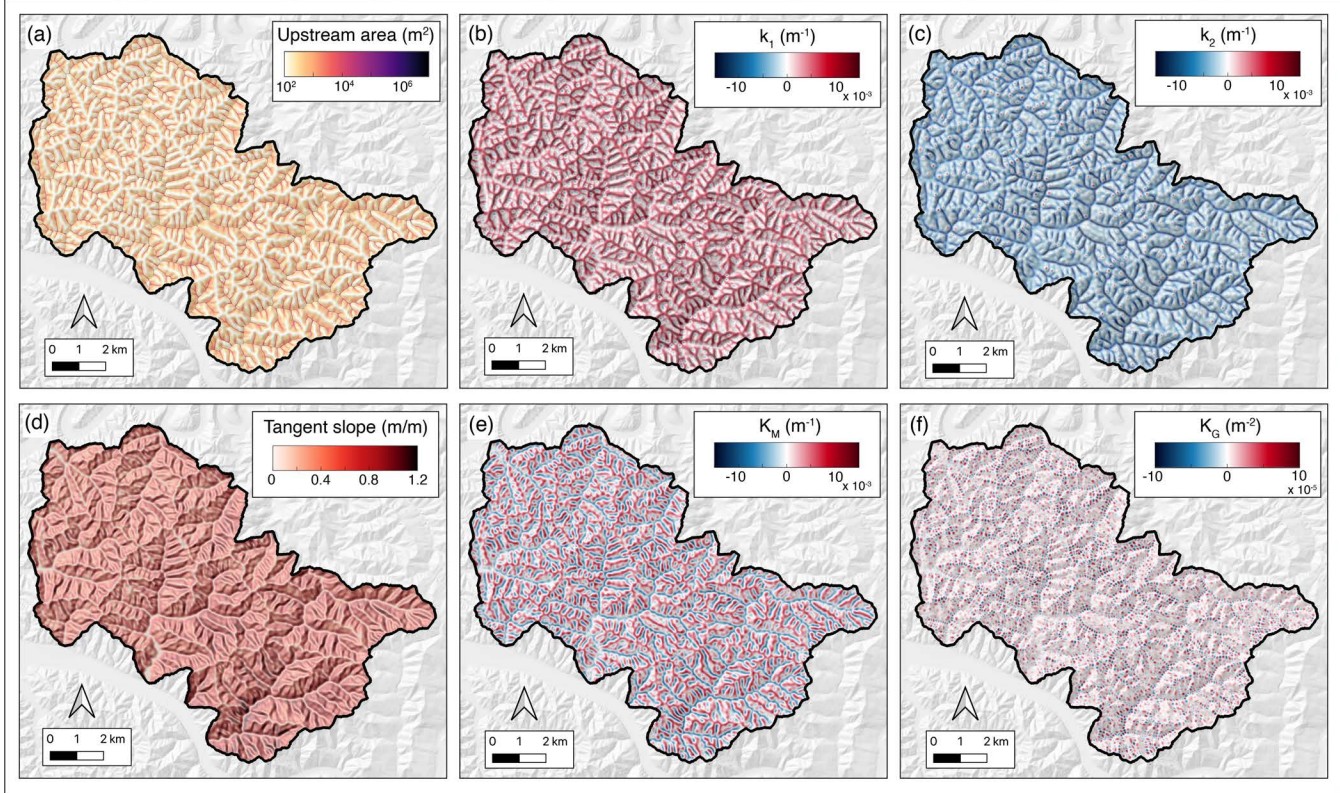

**Figure 6. Map-view distributions of surface geometry metrics. a.** Upstream surface area of grid points calculated with D infinity algorithm and weighting pixels with area-expansion factor $\alpha$. **b.** First principal curvature ($k_1$). **c.** Second principal curvature ($k_2$). **d.** Slope of tangent plane ($S_T$). **e.** Mean curvature ($K_M$). **e.** Gaussian curvature ($K_G$).

To show that this sensitivity of Laplacian curvature to slope persists on topographic surfaces we define a percent error between the $1/2$ Laplacian and $K_M$ on both the synthetic spherical surface and topography of our study area as

$$PE_{LP} = \frac{\nabla^2/2 - K_M}{K_M} \times 100. \tag{39}$$

On the hemisphere $K_M$ is, at every point, the analytically predicted curvature, while on the topographic surface $K_M$ is calculated using the methodology of this paper. Both resulting error grids are binned by tangent slope ($S_T$; Equation 12) with results shown in Fig. 7. c. Median error values for the $1/2$ Laplacian on topography are given by the purple boxes while error on the sphere is given by the red dashed line. We see that fractional errors in Laplacian curvature on the spherical surface are reflected in topography with errors of over $100\%$ when slopes exceed $\sim 1 \; m/m$. This shows that the Laplacian is a poor measure of curvature in steep topography, and that error magnitudes scale with the magnitude of curvature as well as slope. The dependence of Laplacian error on slope can also be seen in a plot of $PE_{LP}$ binned by catchment area (Fig. 7.f) where a prominent spike in error up to $\sim 45\%$ occurs at drainage areas of $\sim 10^3 \; m^2$, which we will see in section 7 encompasses the steepest parts of the landscape (Fig. 9.a).



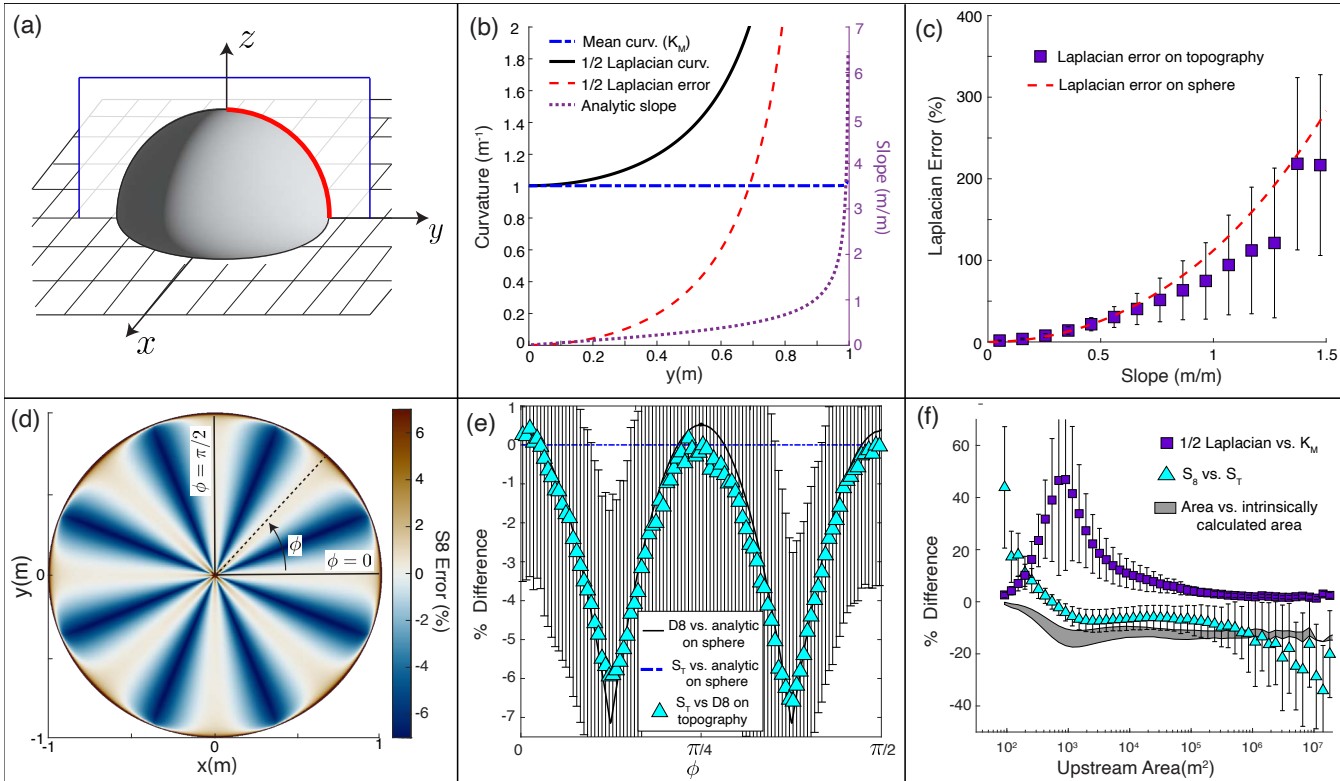

**Figure 7. Comparison of intrinsic surface metrics use in this study with other methods common in the literature. a.** Cartoon depiction of unit hemisphere used for comparison with topographic data. Red line shows curve along which error is evaluated in panel b. **b.** Comparison of Mean curvature ($K_M$) and $1/2$ Laplacian as a function of distance from the origin for plane curve defined by the intersection of the unit hemisphere with the $y - z$ plane. Black line is $1/2$ Laplacian, blue dashed line is Mean curvature calculated using intrinsic method, red dashed line is difference between $1/2$ Laplacian curvature and the curvature of the sphere ($1 \ m^{-1}$), and the purple dashed line is slope of the sphere. **c.** $\%$ error of the $1/2$ Laplacian on the unit hemisphere and $\%$ difference between the $1/2$ Laplacian and Mean curvature on topography binned as a function of slope. Red dashed line is $\%$ error on sphere and purple boxes are median values computed on topography. **d.** $\%$ error of the 8 point connected gradient computed on the unit hemisphere. **e.** $\%$ error of the 8 point connected gradient computed on the unit hemisphere and median $\%$ difference between $S_8$ and $S_T$ as a function of azimuth. **f.** $\%$ difference between our intrinsically calculated topographic metrics and other common methods as a function of drainage area.



## 6.2 Comparison of tangent slope to 8-connected neighborhood gradient

Our approach to computing curvatures requires definition of a unit normal vector at every DEM grid cell, which can be used to define the slope of the tangent plane ($S_T$) via Equation 12. While this method is mathematically equivalent to finding a slope magnitude using the Pythagorean sum of directional derivatives it provides different information than the 8-connected neighborhood gradient method (D8) that is a default slope metric, e.g.,in the geomorphology software packages Topotoolbox (Schwanghart and Scherler, 2014) and LSDTopoTools (Mudd et al., 2019). The D8 method assigns a given pixel the slope between it and its lowest neighbor, an efficient flow routing algorithm (O'Callaghan and Mark, 1984). However its ability to accurately define the slope of a surface depends on both the orientation and complexity of the surface. To address the effects of orientation we apply the TopoToolBox D8 algorithm to the same unit hemisphere as in section 6.1 and calculate the percent error between it and the analytically defined slope ($S$) as

$$PE_{D8} = \frac{D8 - S}{S} \times 100. \tag{40}$$

Figure 7.d shows the result of Equation 40 on a map-view projection of the unit sphere. Error in computed slope varies systematically with orientation of the surface up to magnitudes of $\sim 7\%$ of the slope. In Fig. 7.e this error is binned by azimuth (black line) and compared to both the percent error between $S_T$ and the $S$ (blue dashed line), and the percent difference between $S_T$ and $D8$ on topography, binned by azimuth. The percent error in $S_T$ on the sphere is near zero, while the difference between the various slope metrics on topography tracks the azimuthal trend in D8 error on the sphere. This suggests that similar systematic error arises from D8 slope computations on DEMs. Perhaps non-intuitively, the D8 algorithm tends to underestimate slope if pixels are mis-aligned with the direction of steepest descent.

Finally, we bin the percent difference between $S_T$ and $D8$ by catchment area to track differences in the two metrics through the fluvial network (Fig. 7.f). The highest magnitude errors ($\sim 35\%$) occur on ridges (section 7.1.1), while the next highest magnitude negative errors ($|>20\%|$) occur at the highest drainage areas within the fluvial network (section 7.1.4). We will see in section 7 that this trend tracks that of Mean curvature (Fig. 9.b) suggesting sensitivity of the D8 algorithm to surface curvature as well as orientation.

## 6.3 Catchment surface area versus map-view area

As outlined in sections 3 and 5 the intrinsic formulation of surface geometry is partially motivated by the fact that distances on a sloped surface are greater than distances on their 2-d map representation. This means that pixel dimensions used to compute catchment area in flow routing algorithms underestimate area on the topographic surface by the area expansion factor $\alpha$ defined by Equation 14. To evaluate the effect of projection distortion on catchment area calculations we compute two separate area grids. Using the D-Infinity flow routing algorithm in TopoToolbox (Schwanghart and Scherler, 2014) we calculate a standard extrinsically derived upstream area ($A_E$) using DEM pixel dimensions to assign an area to each grid cell. We also compute the intrinsically derived upstream area ($A$), in which we use the same algorithm but pixel areas are weighted by $\alpha$ to give the true



area of the surface within each pixel. We once again define a percent difference as

$$PE_A = \frac{A_E - A}{A} * 100 \tag{41}$$

and plot the result binned by drainage area in Figure 7.f. Through most of the landscape extrinsic catchment area calculations underestimate catchment surface area by $10 - 15\%$, with potential implications for the interpretation of continuity equations.

# 7 Geometric view of Coast Range topography

As outlined in section 5.3, the Mean and Gaussian curvatures can together classify each DEM pixel uniquely into one of four distinct shape classes. We compute these classifications for our study site, and in this section discuss the resulting distributions

in terms of curvature metrics as function of contributing drainage area. We will argue that this approach effectively maps curvature onto known fluvial erosion process regimes (Montgomery and Foufoula-Georgiou, 1993).

Fig. 8.b shows a probability density function (PDF) for each of the four shape classes as a function of contributing drainage area, which is the probability of a shape class and drainage area value occurring simultaneously ($P(C \cap A)$). Comparison to the drainage area PDF (Fig. 8.a) shows that shape probabilities strongly vary with the likelihood of a given drainage area. For

comparative purposes it is more informative to calculate the conditional probability, which we do by invoking the probability axiom (Dekking et al., 2005)

$$P(C \cap A) = P(C|A)P(A) \rightarrow P(C|A) = \frac{P(C \cap A)}{P(A)}, \tag{42}$$

where $P(A)$ is the probability of pixel having a given upstream area. $P(C|A)$ is the conditional probability (Fig. 8.c), or probability of a given shape class occurring at a point with a given upstream area (**?**). This can be used to demonstrate how

land surface geometry records process transitions in fluvial landscapes.

Fig. 9 shows the the distributions of slope, area, Mean and Gaussian curvatures, and surface shape classes in area-space, as-well-as the distribution of shape classes in map view. In Fig. 9.d we see that the landscape is evenly partitioned into concave up and concave down geometric structures implying a landscape organizational signal reflected in Mean curvature that will be discussed further in section 7.2.

## 7.1 Landscape partitioning from Gaussian curvature

Noting significant and systematic variation in shape class distributions and curvature metrics with drainage area (Fig. 9.a-c), we demonstrate a landscape segmentation approach using inflection points (zero crossings) of the two curvature invariants. This approach is motivated by physical theory, as the signs of both $K_M$ and $K_G$ have implications for mass transport phenomena. The sign of $K_M$ records the divergence versus convergence of local gradients, while the sign of $K_G$ differentiates between

stable and unstable 'critical points' with theoretical implications for how the surface respond to disturbances (Goldsten et al., 2002; Matsumoto, 2001). As our partitioning approach is geometrically motivated we choose a labeling scheme $\Sigma_i^j$ based



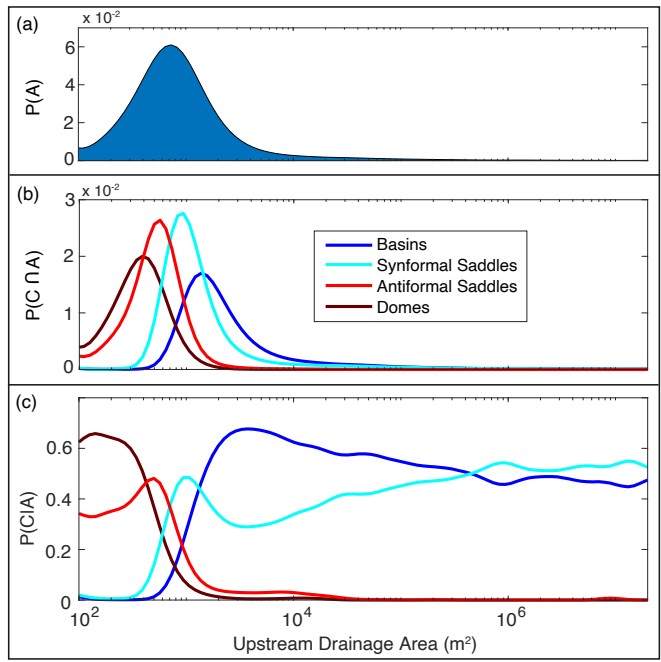

**Figure 8.** Shape class distributions as a function of drainage area. **a**. Probability density function of upstream drainage areas within the study area. **b**. Probability density functions of shape classes as a function of drainage area. **c**. Conditional PDFs of shape classes at a given drainage area.

solely on geometry. Subscripts indicate the curvature invariant used ($i = G$ for $K_G$ and $i = M$ for $K_M$), while superscripts $j$ correspond to the number of previous zero crossings in area space ($j = 0$ corresponds to the zero crossing at smallest drainage area). This provides a convenient reference for curvature domains expressed as portions of a landscape, and could facilitate

future comparative studies that explore differences in the relationship between geometry and process across diverse tectonic, climatic, and lithologic settings.

### 7.1.1  $\Sigma_G^0$: Drainage areas less than $3.75 \times 10^2$ m$^2$

In fluvial landscapes the smallest drainage areas are associated with ridge-peak networks that separate neighboring watersheds (Scherler and Schwanghart, 2019). We define a landscape region (that we call $\Sigma_G^0$) containing all land surface points with

drainage areas less than $3.75 \times 10^2$ m$^2$, a value coincident with the first area-space inflection in Gaussian curvature (Fig. 10.b). In this region both the Mean and Gaussian curvatures are dominantly positive reflecting downward concavity of topography and divergence of surface gradient vectors (O'Neill, 2006). This is consistent with the idea that ridge networks lack convergent overland flow (Fenneman, 1908), and so mass transport is accomplished through diffusive processes with rates that scale with curvature and site-specific diffusivity (Roering et al., 1999). Thus high magnitude positive Mean curvatures in this region

indicate high rates of diffusive mass transport, likely required for erosion along ridge lines to keep pace with advective mass



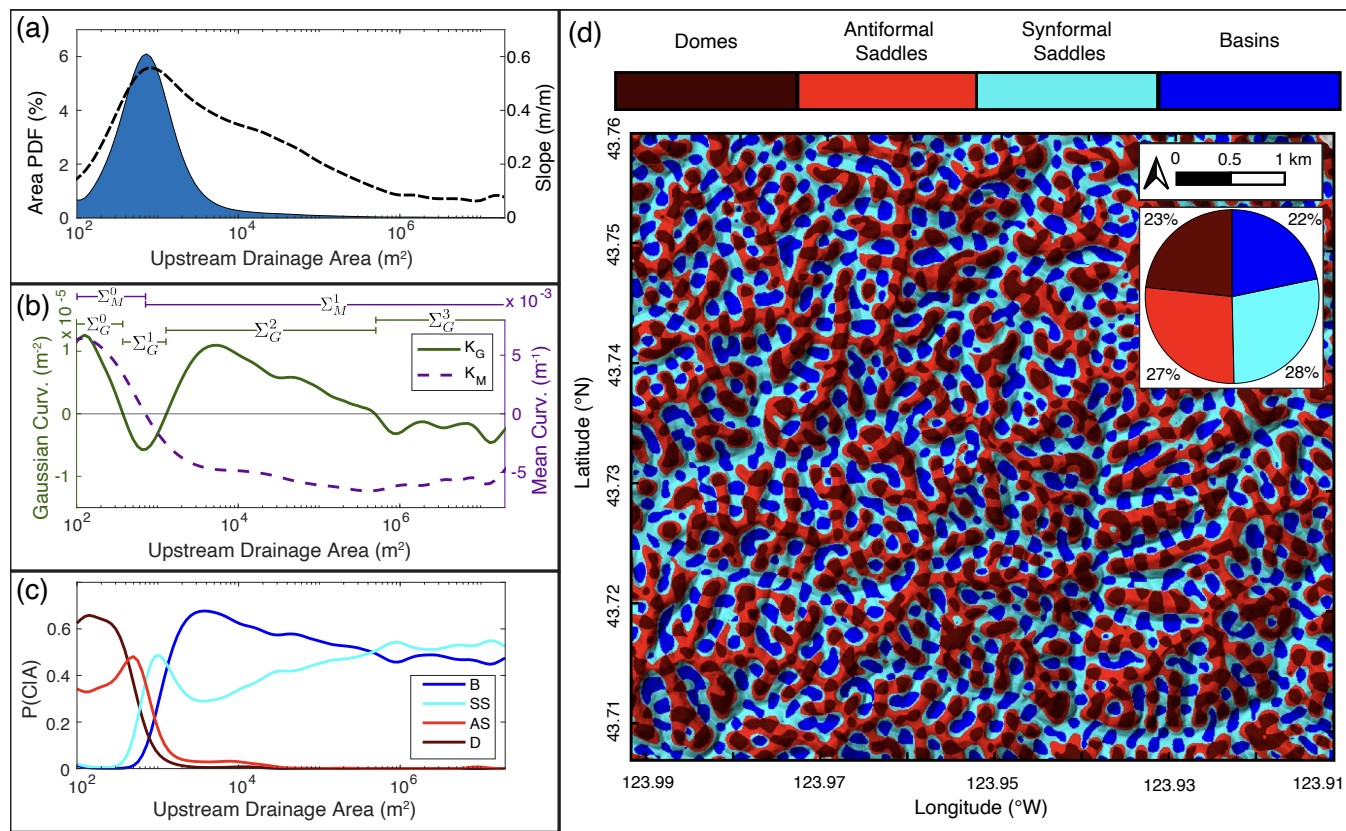

**Figure 9.** Distribution of derived surface geometry metrics computed on the full region of interest. **a)** PDF of upstream drainage areas. **b)** Gaussian and mean curvatures binned by upstream drainage area. Horizontal lines at top of panel show $\Sigma$ regions outlined in sections 7.1 and 7.2. **c)** Conditional PDFs of shape classes as a function of upstream drainage area. **d)** Map of shape classes projected on a focused subregion of the study area. Pie chart inset shows shape-class composition of the surface.

transport in channel networks below. This idea is supported by well documented correlations between Laplacian hilltop curvature and catchment-scale erosion rates in the Oregon Coast Range, where curvature is usually calculated as the Laplacian operator applied to isolated hilltop regions (Struble et al., 2024). Our approach allows for physics-informed delineation of this diffusion-dominated region, while robustness of our approach against systematic slope and curvature dependent error (section

6) allows for continuity in topographic geometry calculations between this high curvature region and the steeper hillslope domain below (Roering et al., 2001a).

The map-view representation of the $\Sigma_G^0$ region (Fig. 10.d) shows the spatial distribution of diffusion dominated regions with implications for the organizational structure of fluvial landscapes. According to this partitioning scheme, the ridge-peak network makes up 19% of the land surface area, with 63% of surface points making up localized dome structures (peaks) that

are connected by antiformal saddles (ridges) that comprise the remaining 37% (Fig. 10.a,d). Transitions between these shape



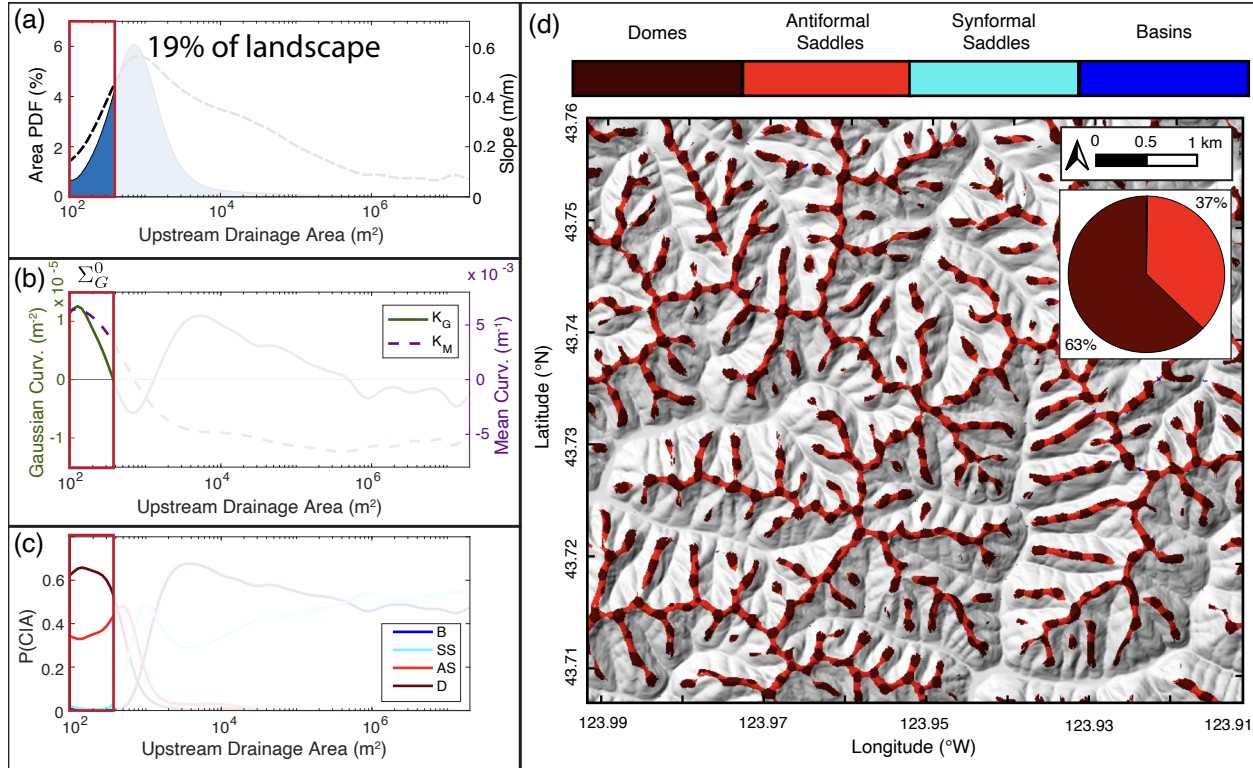

**Figure 10. Surface geometry data for points in the landscape with upstream drainage areas less than $3.75 \times 10^2$ m$^2$.** The red box in each of the area-binned plots (a-c) highlights the range of included drainage areas. **a)** PDF of upstream drainage areas. **b)** Gaussian and Mean curvatures binned by upstream drainage area. **c)** Conditional PDFs of shape classes as a function of upstream drainage area. **d)** Map of shape classes projected on a focused subregion of the study area. Pie chart inset shows shape-class composition of the surface.

classes along ridge lines show oscillations between positive and negative Gaussian curvatures, analogous to the alternating "summits" and "knots" of Cayley (1859), and the "hills" and "passes" of Maxwell (1870). We will elaborate on this connection to early landscape organization theories in section 7.3.

### 7.1.2 $\Sigma_G^1$: Drainage areas between $3.75 \times 10^2$ m$^2$ and $1.29 \times 10^3$ m$^2$

As drainage area increases beyond $375$ m$^2$ the binned Gaussian curvature changes sign and remains negative up to drainage areas of $1290$ m$^2$ (Fig. 11.b). This region ($\Sigma_G^1$) makes up the majority of the landscape with $54\%$ of pixels falling in this relatively narrow range of catchment areas. This is also the steepest landscape region (Fig. 11.a) coinciding with hillslopes where loose material moves downhill through a combination of gradient driven processes such as landsliding, granular creep, and stochastic particle dislocation (Roering et al., 2001b; Jaeger and Nagel, 1992; Furbish et al., 2009; Deshpande et al., 2021; 520 Gabet, 2003).



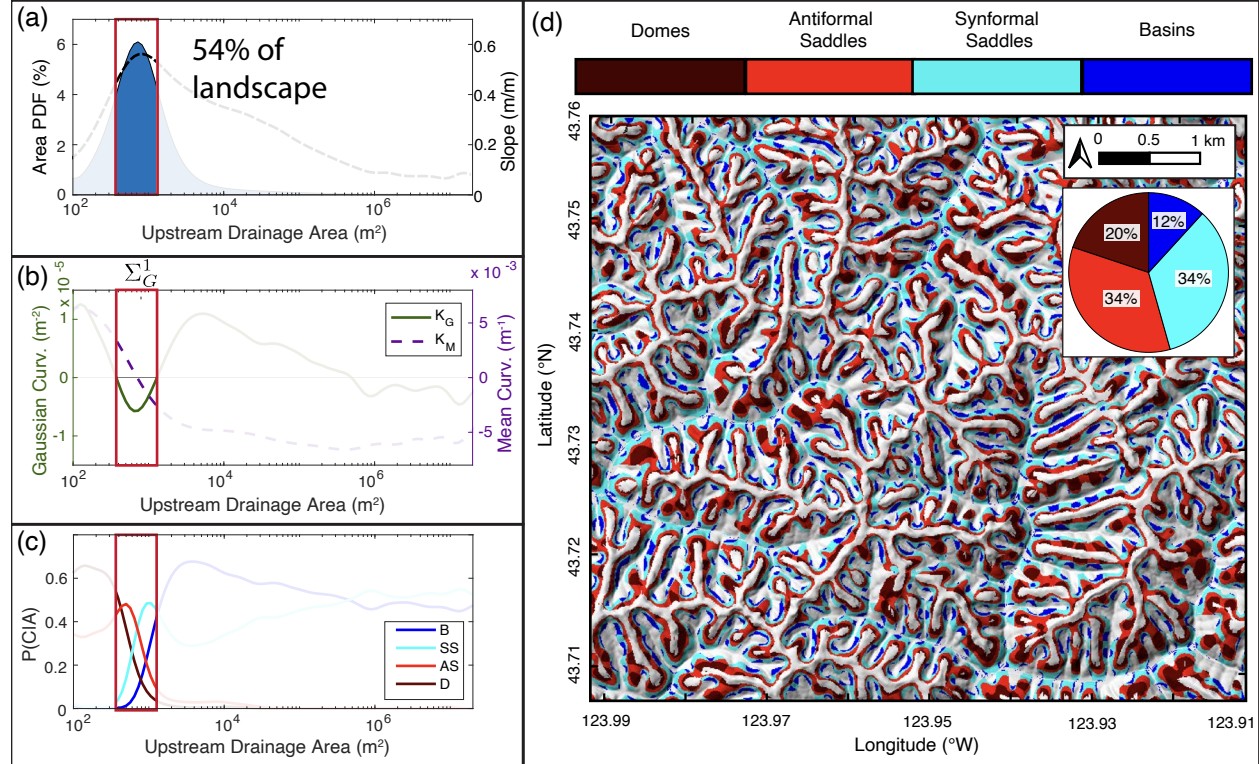

**Figure 11. Surface geometry data for points in the landscape with upstream drainage areas between $3.75 \times 10^2$ m$^2$ and $1.29 \times 10^3$ m$^2$.** The red box in each of the area-binned plots (a-c) highlights the range of included drainage areas. **a)** PDF of upstream drainage areas. **b)** Gaussian and Mean curvatures binned by upstream drainage area. **c)** Conditional PDFs of shape classes as a function of upstream drainage area. **d)** Map of shape classes projected on a focused subregion of the study area. Pie chart inset shows shape-class composition of the surface.

Here the landscape undergoes a major concavity transition between almost uniformly divergent topography ($\Sigma_G^0$; section 7.1.2) and basins at the head of the convergent drainage network (Dietrich et al., 1993). This region contains the Mean curvature's only area-space inflection point at 740 m$^2$, which is aligned with the global minimum in binned Gaussian curvature and global maximum in topographic slope (Fig. 11.a-b). Geometrically, this transition is seen in the rapidly evolving shape-class

distributions in Fig. 11.c which suggest a high level of surface complexity across this concavity transition; complexity that is not captured by analyses of hillslopes conducted along 1-d profiles (Roering et al., 2001a), but perhaps is better captured by approaches that leverage contour curvature (Bonetti et al., 2018). In section 7.2 we will show a landscape partitioning scheme based on this concavity inflection, which may itself represent a strong landscape organization signal.

### 7.1.3    $\Sigma_G^2$: Drainage areas between $1.29 \times 10^3$ m$^2$ and $3.30 \times 10^5$ m$^2$

At drainage areas of $1.29 \times 10^3$ m$^2$ the binned Gaussian curvature again changes sign, increasing to a local maximum at $\sim 4500$ m$^2$ before steadily decreasing back to zero at $3.30 \times 10^5$ m$^2$ (Fig. 12.b). We define our third landscape region ($\Sigma_G^3$) between





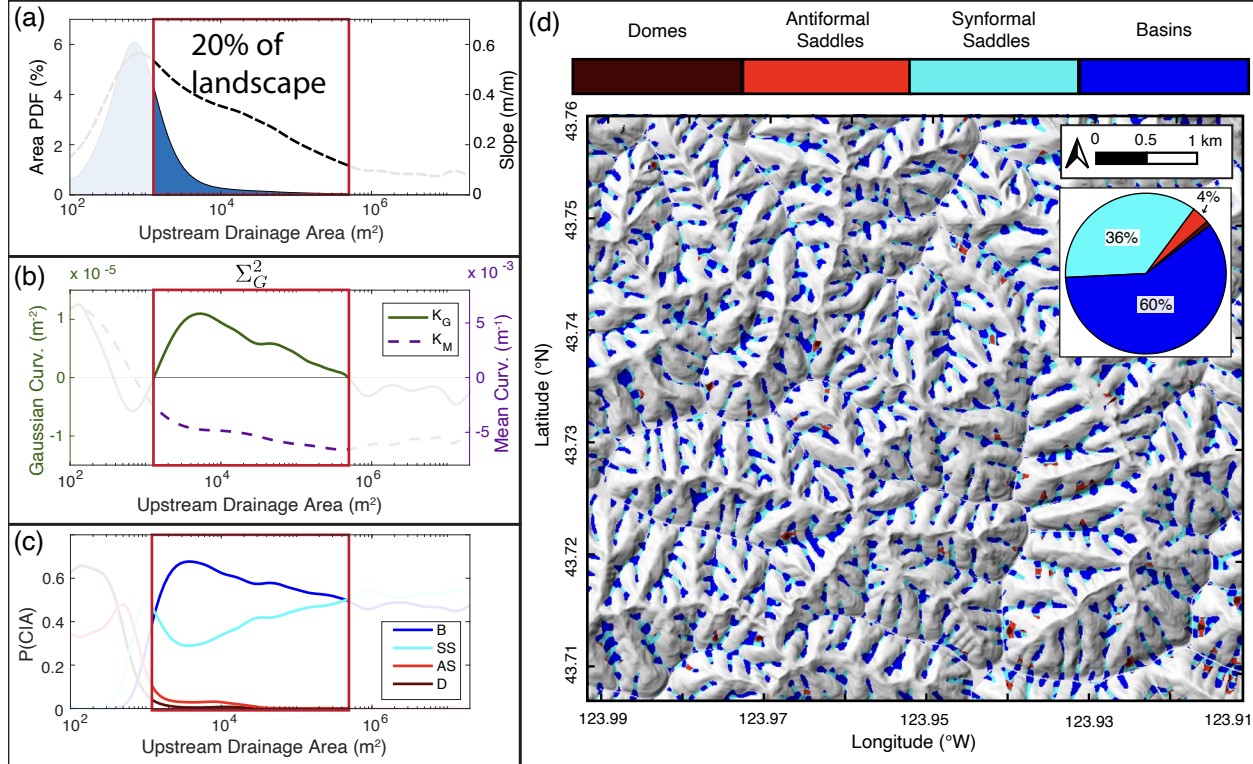

**Figure 12. Surface geometry data for points in the landscape with upstream drainage areas between** $1.29 \times 10^3$ **m$^2$ and** $3.30 \times 10^5$ **m$^2$.**
The red box in each of the area plots highlights the region of interest. **a)** PDF of upstream drainage areas. **b)** Gaussian and mean curvatures binned by upstream drainage area. **c)** Conditional PDFs of shape classes as a function of upstream drainage area. **d)** Map of shape classes projected on a focused subregion of the study area. Pie chart inset shows shape-class composition of the surface.

these inflection points. Here, convergence of surface gradient vectors is indicated by negative $K_M$ and the dominance of basins (60%) and synformal saddles (36%). In map view (Fig. 12.d) the existence of basins at the smallest drainage areas are likely the geometric expression of colluvial hollows at the head of debris-flow networks (Dietrich et al., 1993). As drainage area

increases past the maximum in Gaussian curvature the consistent negative slope in both the Gaussian and Mean curvatures is consistent with increasing downstream channelization in debris-flow channels (Stock and Dietrich, 2003; Mcguire et al., 2022). This same trend is apparent in the shape class distributions in Fig. 12.c where a decrease in basin points trades off with an increase in synformal saddles as the landscape becomes increasingly convergent.

This region makes up 20% of the study area, similar to the 19% of the landscape encapsulated by the other region of positive

Gaussian curvature ($\Sigma_G^0$). In addition, the percent composition of basins (60%) and synformal saddles (36%) roughly mirrors the $\Sigma_G^0$ composition of domes (63%) and antiformal saddles (37%). This symmetry in curvatures will be discussed further in section 7.2.



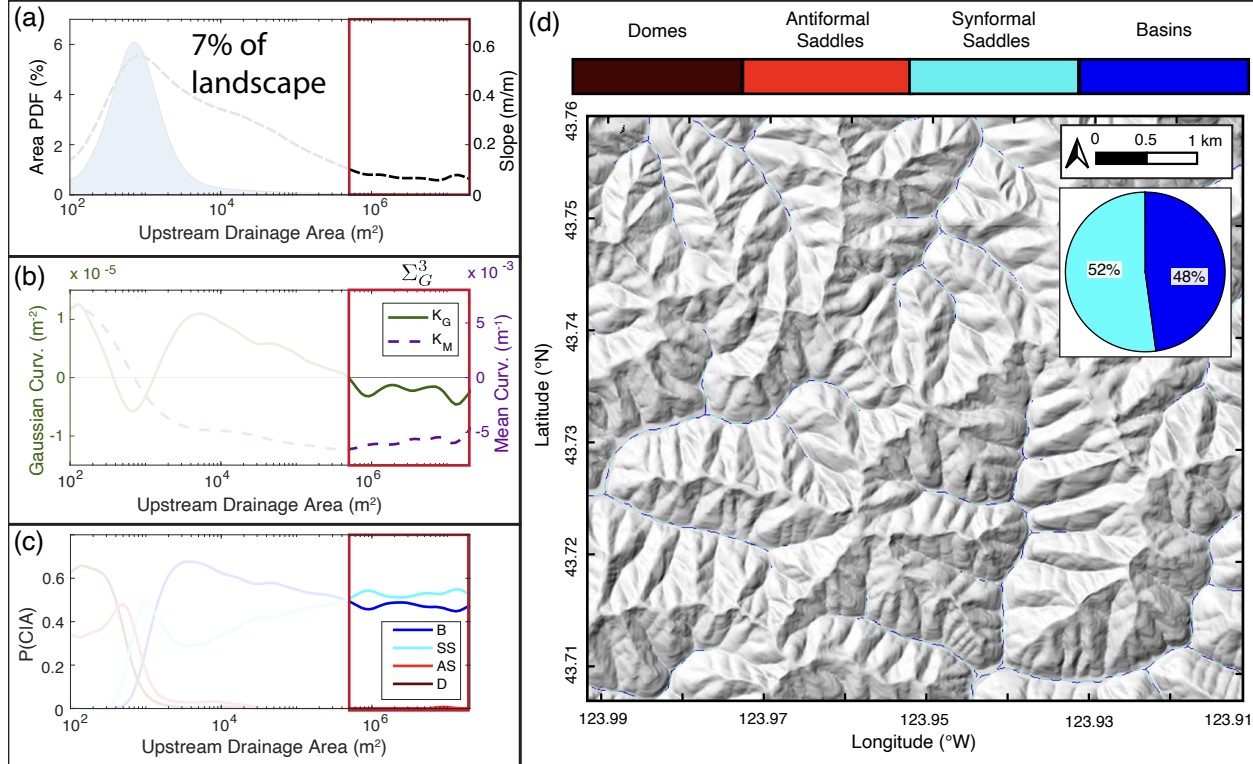

**Figure 13. Surface geometry data for points in the landscape with upstream drainage areas between** $4.5 \times 10^3$ **m$^2$ and** $9.8 \times 10^5$ **m$^2$.** The red box in each of the area plots highlights the region of interest. **a)** PDF of upstream drainage areas. **b)** Gaussian and Mean curvatures binned by upstream drainage area. **c)** Conditional PDFs of shape classes as a function of upstream drainage area. **d)** Map of shape classes projected on a focused subregion of the study area. Pie chart inset shows shape-class composition of the surface.

### 7.1.4 $\Sigma_G^3$: Drainage areas greater than $3.30 \times 10^5$ m$^2$

The last inflection point in binned Gaussian curvature occurs at drainage areas of $3.30 \times 10^5$ m$^2$, where synformal saddles
surpass basins as the dominant morphology (Fig. 13.c). This intuitively implies a growing influence of channels in defining landscape curvatures, and so we associate this final region ($\Sigma_G^3$) with the fluvial network, noting that drainage areas between $10^5$ and $10^6$ m$^2$ are commonly associated with the onset of fluvial processes (Montgomery and Foufoula-Georgiou, 1993). In terms of surface area the contribution of this region is small (7%; Fig. 13.a) with little geometric change across the two orders of magnitude spanned by drainage area. The only overall trend is a gradual increase in Mean curvature, which is consistent
with downstream valley widening as erosional efficiency of river channels increase with drainage area. However, a detailed look at the map-view shape distribution (Fig. 13.d) reveals regular transitions between basin and saddle structures, indicating along-channel oscillations in the first principal curvature ($k_2$ is always negative in a channel) that we will revisit and discuss in section 7.3.





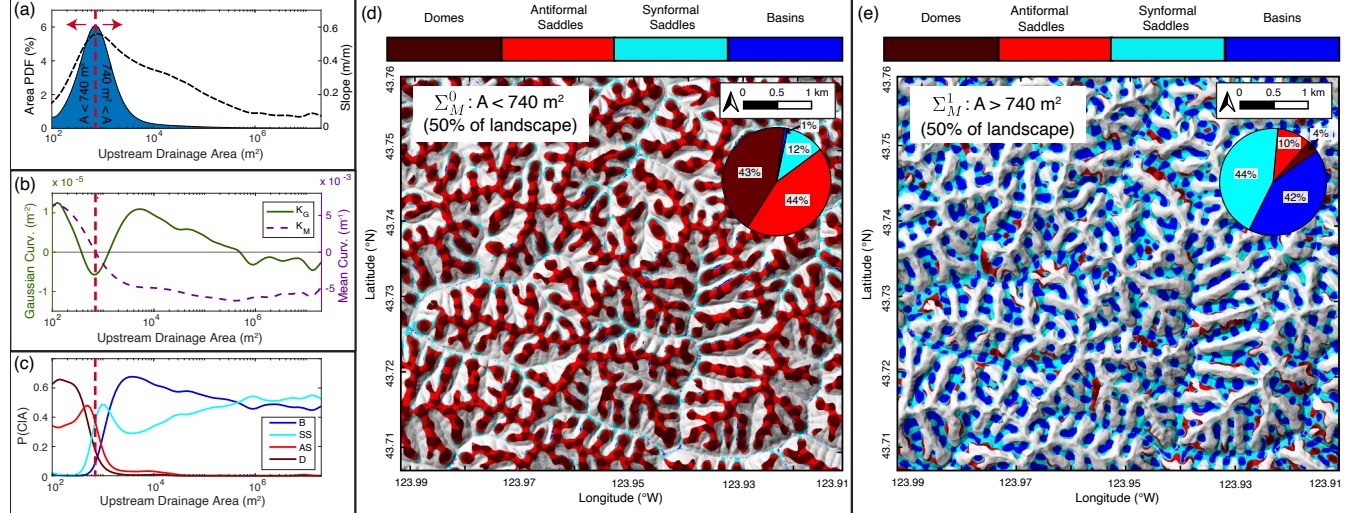

**Figure 14. Maps of surface geometry for landscape partitioning about the Mean curvature inflection point at drainage areas of** 740 **m$^2$. a.** PDF of upstream drainage area. Dashed red line shows point of landscape partitioning. **b)** Gaussian and mean curvatures binned by upstream drainage area. **c)** Conditional PDFs of shape classes as a function of upstream drainage area. **d)** Map of shape classes for areas of the landscape with drainage areas greater than 740 m$^2$. Pie chart inset shows shape-class composition of the surface. **e)** Map of shape classes for areas of the landscape with drainage areas smaller than 740 m$^2$. Pie chart inset shows distribution of shape classes.

All of the process regimes identified in this Gaussian curvature decomposition have been previously identified in the literature
(Montgomery and Buffington, 1997). However it is remarkable that a single topographic metric can capture the full range of process transitions in a fluvial landscape. Tracking the Gaussian and Mean curvatures simultaneously through area-space suggests ways to constrain geomorphic processes in transitional portions of landscape that are difficult to characterize with extrinsically derived geometry metrics.

### 7.2  Landscape partitioning from Mean curvature

Section 7.1 demonstrates that Gaussian curvature inflection points effectively classify landscape process transitions. In this section we show that Mean curvature also provides a useful segmentation scheme. We decompose the landscape into two regions ($\Sigma_M^0$ and $\Sigma_M^1$) separated by the single inflection in $K_M$ at drainage areas of 740 m$^2$. Results are shown in Fig. 14. Alignment between this curvature inflection point and the peak of the slope curve in Fig. 14.a is consistent with the idea that curvature decreases as hillslope profiles approach an angle-of-repose above which slope are gravitationally unstable Roering
et al. (2007). Downhill of this point, slope decreases and unconsolidated material will tend to collect as colluvium at the head of the channel network (region $\Sigma_G^2$).

The landscape is equally distributed about this zero crossing such that 50% of points are above/below the most probable drainage area value in our study area. Partitioning the landscape this way reveals surprising symmetries in both shape class





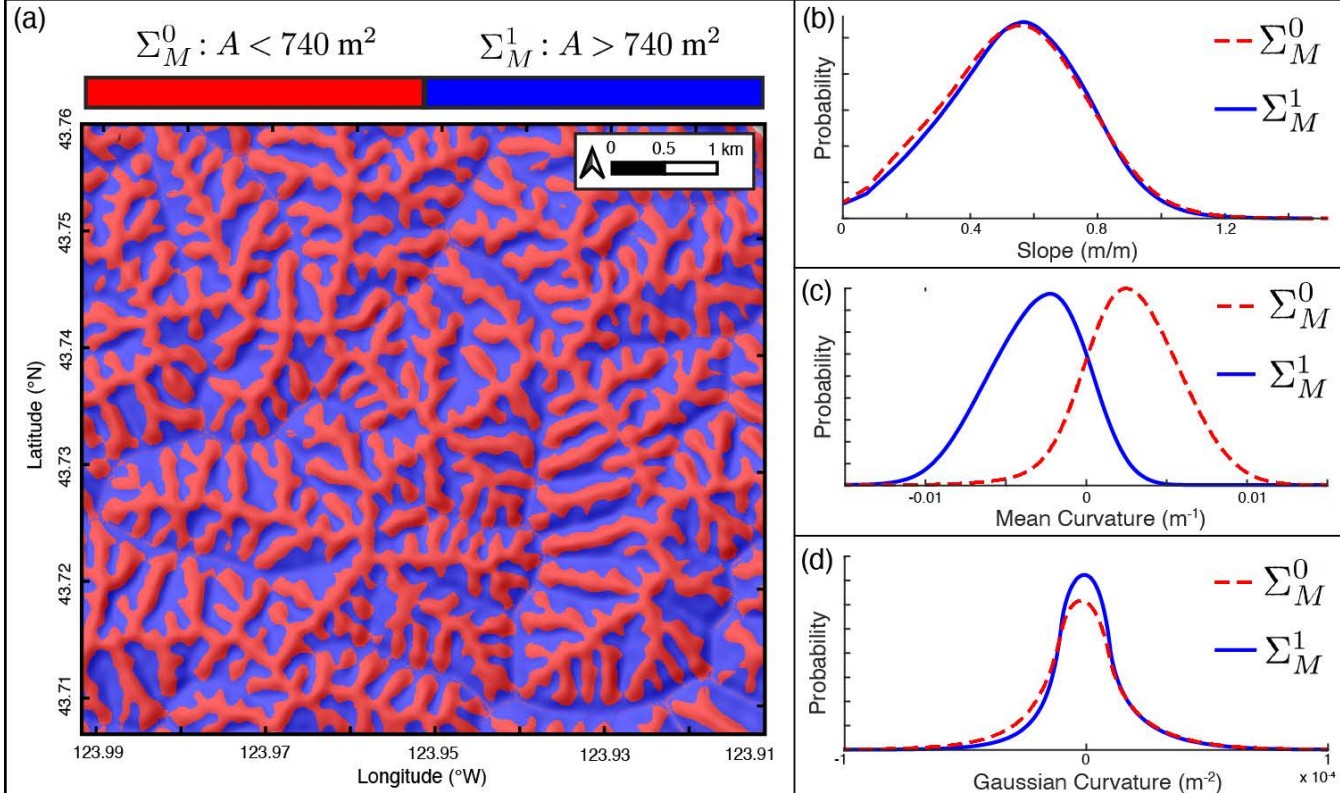

**Figure 15. Distribution of surface geometry metrics for regions defined by the Mean curvature inflection point. a.** Map-view of landscape partitioned about the point of inflection in $K_M$. **b.** Distribution of tangent slope in regions of both negative and positive average $K_M$. **c.** Distribution of Mean curvature in regions of both negative and positive average $K_M$. **d.** Distribution of Gaussian curvatures in regions of both negative and positive average $K_M$

distributions and surface geometry metrics of the two regions (Fig. 14.d-e). Figure 15.a shows a map of the study area divided into concave and convex domains based on this area threshold. Probability distributions of slope, Mean curvature, and Gaussian curvature for the two regions are shown in Fig. 15.b-d. While the slope and Gaussian curvature are similarly distributed in the concave and convex landscape regions, we see that the Mean curvature is distributed symmetrically such that the total Mean curvature in the landscape is close to zero. This observation could be interpreted as reflecting a "Minimal Surface" condition (Han and Che, 2018) for steady state landscapes, in which total Mean curvature is minimized. Making quantitative process-based connections to Minimal Surface theory is beyond the scope of this paper, however.

## 7.3 Geometric properties of channels and ridges

We have thus far focused on documenting Oregon Coast Range landscape segmentation in drainage area from a curvature perspective. A clear corollary to this is to ask specifically about the emergent channel and ridge network structures that manifest





from this drainage area segmentation. It is well established that curvature provides a powerful tool for extracting continuous

concave-up structures and deriving definitions of channel networks that are self-consistent throughout the landscape (Passalac-qua et al., 2010; Gallant and Hutchinson, 2011; Bonetti et al., 2018). Our methods are suitable for this task as well, and for the parallel extraction of concave-down ridge network structures (Scherler and Schwanghart, 2019), but we will not pursue that objective here.

Instead, we will focus on the strikingly even partitioning of Mean curvature between concave up structures (channels) and

concave down structures (ridges). These structures are themselves composed entirely of alternating basins and synformal saddles (in channels) and domes and antiformal saddles (on ridges). Figure 16.a shows a close-up of our study area around Franklin Creek to demonstrate this pattern. While the size distribution of these alternating shape classes within a channel or ridge is variably sensitive to lowpass filter threshold, the shape classes themselves are much more robust as they reflect zero crossings in $K_M$ and $K_G$ whose positions are insensitive to filter cutoff (Fig. 5). This alternating pattern of local shape classes,

originally recognized qualitatively by Cayley (1859) and Maxwell (1870) seems to be a fundamental aspect of channel and ridge network geometry.

Fig. 16.b-c plots in blue the elevation of Franklin Creek and its south ridge as a function of distance from the most downstream point of the creek (where it intersects the Umpqua river). The drainage area (red curves) along these structures (calculated using the intrinsic area calculation method in section 6.3) reflects expectations: discontinuities in drainage area along the channel

correspond to tributary junctions while ridge-top drainage area deviates from one grid cell only in saddles (up to 8 grid cells long here) between local maxima.

Fig. 16.d-e plots the signed principle curvatures for ridge and channel. An immediate comparison of note is that local minima in $k_1$ for the channel and $k_2$ for the ridge correspond to basins and antiformal saddles respectively (circles). These structures align with tributary junctions in the channel, and lie directly upslope of 1st order channel heads on the ridge. This indicates that

the curvature shape classes reflect structural changes in network geometry for both concave and convex topography. Neither structure – the local basins at channel junctions or saddles on ridgetops corresponding to transitions from hillslopes to channels – have been previously described to our knowledge.

Fig. 16.f-g then plots the Gaussian ($K_G$) and Mean ($K_M$) curvatures along the channel and ridge. The notable comparison in this case is that local maxima in $K_G$ and $K_M$ are anticorrelated along the channel and correlated along the ridge. This

symmetry reflects the paired shape classes in either structure.

In comparing channel and ridge geometries we see that, in both cases, the two measures of curvature (principle curvatures or invariants) are near mirror-images of each other. In both cases one metric oscillates around zero (a principle curvature in 16.d-e and $K_G$ in 16.f-g), while the other is strictly positive (for ridge) or negative (for channel), although still oscillatory. The along-channel and along-ridge envelope of this latter metric varies non-monotonically, as to be expected for a small dendritic

drainage basin, but nonetheless exhibits coincident channel widening (decrease in the magnitude of $k_2$) and ridge narrowing (increase in $k_1$ towards the mouth of Franklin Creek (distances $\lesssim 2300$ m).





Finally, while physics-based modeling of curvature is outside the scope of this work, it is informative to compare the observed curvature of channel and ridge to theoretical models. Fig. 16.c-E show best fitting powerlaws to ridge and channel, after (Whipple and Tucker, 1999) for bedrock channel longitudinal profile and models such as those of Willett (2010) for interfluvial ridge elevations. The fit constants are $a = 6$ m, $b_1 = 2.84 \times 10^{-7}$ m$^{-1.4}$, $c_1 = -1.4$, $b_2 = 41.88$ m$^{0.72}$, $c_2 = 0.28$.

For Franklin Creek, while the elevation profile is well-fit by a stream power-law fit, the resulting curvature (obtained by differentiating the longitudinal profile twice) does not capture oscillations observed in $k_1$, the along-channel principal curvature. And yet the average value of the stream power model curvature ($1.1 \times 10^{-5}$ m$^{-1}$) is close to the average value of $k_1$ ($8.3 \times 10^{-5}$ m$^{-1}$) extracted from the DEM (we expect an even closer match if tributaries are included in the stream power model, e.g., Willett (2010)). Thus, the steady state model approximates the average concavity of the true channel geometry, despite much larger curvature oscillations associated with local basin structures at tributary junctions.

Similarly, a power-law fit to the Franklin Creek south ridge profile in Fig. 16.c well represents the elevation but misses the smaller scale curvature oscillations oscillations between domes and antiformal saddles. The average value of this fit ($-3.3 \times 10^{-4}$ m$^{-1}$) is with 7% of the average value of $k_2$ ($-3.5 \times 10^{-4}$ m$^{-1}$) extracted from the DEM, reflecting the overall concave down nature of the along-ridge curvature. These results suggest that standard fluvial process models, while missing physical ingredients at smaller scale, capture network-scale curvatures of channels and ridges.

# 8 Future directions

Quantitative classification of landforms and topography generally is challenged by the myriad interacting physical processes shaping landscapes at a range of spatial and temporal scales. Nevertheless, certain geomorphic metrics such as local slope and upstream drainage area have, through extensive empirical validation, proven to be useful indicators of spatial process transitions (Montgomery and Foufoula-Georgiou, 1993; Rosenbloom and Anderson, 1994; Stock and Dietrich, 2003) and transient landscape evolution (Kirby and Whipple, 2012; Royden and Perron, 2013).

While it is premature to claim that the curvature invariants used here have similar broad utility across landscapes, in our Coast Range study area these metrics – referenced to drainage area through $\Sigma_G^j$ and $\Sigma_M^j$ thresholds (Figs. 10-15) – separate the landscape into regimes that can be clearly associated with well known geomorphic processes. The $\Sigma_G^j$, $\Sigma_M^j$ are separated by area-space inflection points (zero crossings) in Gaussian and Mean curvature and appear to be minimally sensitive to DEM quality or smoothing. These $\Sigma_i^j$ regimes, reflecting areas dominated by different combinations of convex and concave shape classes, should occur in all landscapes because they encode a distribution of 'critical points' that characterize stability and continuity in all 2D surfaces (Matsumoto, 2001; Goldsten et al., 2002). These geometries have implications for the sensitivity of landforms to external perturbation (Goldsten et al., 2002), and so we expect that comparison of the drainage area values associated with $\Sigma_i^j$ domains – perhaps in particular the concavity transition between $\Sigma_M^0$ and $\Sigma_M^1$ – to reflect signatures of landscape disequilibrium.







**Figure 16. Characteristics of Franklin Creek and its south ridge.** (a) Curvature shape classes with channel and ridge highlighted in yellow and magenta. Circles are local minima of along-channel principal curvature $k_1$ while squares are local minima of along-ridge principal curvature $k_2$. (b) Ridge elevation profile (left axis) and drainage area (right axis). Dashed line is drainage area for one cell. Pink curve is a powerlaw fit (fit parameters are listed in text). (c) Channel elevation profile (left axis) and drainage area (right axis). Pink curve is a powerlaw fit. (d) Principal curvatures along the south ridge. Note that local minima in $k_2$ (black circles) correspond to local saddles directly upslope from 1st order channel heads in (a). The mean of $k_1 = 0.01$ m$^{-1}$ and the mean of $k_2 = -3.49 \times 10^{-4}$ m$^{-1}$. (e) Principal curvatures along Franklin Creek. Note that local minima in $k_1$ (black circles) correspond to junctions between tributary channels in (a). The mean of $k_1 = 8.32 \times 10^{-5}$ m$^{-1}$ and the mean of $k_2 = -0.01$ m$^{-1}$. The red curve comes from stream power (it has a mean value of $1.10 \times 10^{-5}$ m$^{-1}$). (f) and (g) Profiles of Gaussian curvature $K_G$ and Mean curvature $K_M$ along ridge and channel.



More broadly, the presence of persistent curvature patterns in structures of interest suggest the potential for new insights into geomorphic processes. For example, oscillations in principal curvatures within fluvial channels ($\Sigma_G^3$ and Fig. 16) capture a step-pool morphology that is well documented in field studies (Grant et al., 1990; Montgomery and Buffington, 1997; Chartrand et al., 2011). This channel morphology is not explained by landscape evolution studies that equate erosion rates to in-channel shear stress averaged over large spatio-temporal scales (Whipple and Tucker, 1999), but we show here that these oscillations are first order features of fluvial channel networks in the Oregon Coast Range. While the magnitude of these oscillations in curvature would need to be validated by field studies before any quantitative connections to process could be defined, the ability to potentially detect step-pool morphology at the landscape scale could open the door to connecting localized models of mass transport in rivers to landscape scale erosion models applied to topographic datasets (Venditti et al., 2020; Church and Zimmermann, 2007; Scheingross and Lamb, 2017; González et al., 2017; Escauriaza et al., 2023).

Similarly, the ability to robustly identify colluvial hollows (a prominent component of $\Sigma_G^2$), where the topographic surface is shaped by a superposition of competing processes at the onset of convergent topography (Dietrich et al., 1993), illustrates the power of this approach. In landscape regions strongly shaped by debris flow processes (Struble et al., 2023), strongly disequilibrium dynamics (Donahue et al., 2013), glacial erosion (Kober et al., 2019), or even those dominated by constructional landforms such as in volcanic terrane (Karlstrom et al., 2025), slope-area scaling and other commonly used process-oriented classification approaches break down and tools such as developed here are likely to be useful. Because surface curvature also influences shallow subsurface stress state for rock fracture (Martel, 2011; Clair et al., 2015; Moon et al., 2017) and the hydraulic gradients driving groundwater flow (Wörman et al., 2006; Zhang et al., 2022), we expect that problems in Critical Zone science may also be examined through the lens of topographic curvature (Riebe et al., 2017).

Many processes driving landscape evolution have an intrinsic scale length (Wegmann et al., 2007; Crozier et al., 2018; Roering et al., 2010), so the combination of careful spectral filtering to isolate certain topographic features with curvature analysis seems a promising direction for future efforts in complex geomorphic settings (Perron et al., 2008; Struble et al., 2021). For example, 1-d measures of hillslope length in the Oregon Coast Range (Grieve et al., 2016; Roering et al., 2007) could be compared to average path lengths in the $\Sigma_G^1$ region to quantify similarities between intrinsic and extrinsic approaches, with a physically justified definition of the domain boundary given by our partitioning scheme. Quantitative comparison of our results with such geometric studies that focus on isolated landscape domains is a clear next step in development of these methods.

From a practical standpoint, Fig. 7.f highlights how intrinsic geometric computation of topographic metrics such as slope, curvature, and upstream drainage area differ from the standard approach using an extrinsic map view projection of a DEM. The $\Sigma_i^j$ regimes (e.g., as illustrated on Fig. 9.b) appear to be relevant. For example, curvature and drainage area computed over $\Sigma_G^1$, encompassing steep hillslopes, exhibit average errors of $\sim 50\%$ and $\sim 15\%$ respectively which are larger than any other segment of the landscape. Slopes computed in either $\Sigma_G^0$ or $\Sigma_G^3$ are maximally different by $\sim 30\%$, representing the smallest and largest drainage areas. Because the $\Sigma_G^1$ region accounts for the majority of land surface area (Fig. 11.a), error in drainage area from $\Sigma_G^1$ persists across all higher drainage areas with error values of $\sim 10\%$. Understanding the effects of projection distortion on empirical scaling relations (e.g., Hack's Law, Hack et al. (1957), relies on drainage area computed

from a DEM), and process-based models (e.g., sediment mass-continuity depends on local curvature, stream power relies on slope and drainage area, Whipple and Tucker (1999)), is left for future work, but we suspect it may be non-negligible in some applications.

## 9 Conclusions

In this work we have shown how the intrinsic surface characterization approach of Carl Frederick Gauss provides a powerful framework for deriving topographic geometry metrics for landform characterization and landscape segmentation. We have shown that common metrics of topographic form, such as slope, upstream drainage area, and Laplacian-based curvature at a point are accurately captured in an intrinsic coordinate system. This approach allows us to show that errors in Laplacian curvatures are sensitive to slope and curvature, D8 slope methods are sensitive to orientation and curvature, and stream catchment area measured on the topographic surface differs by around 10% from area calculated on a map-view projection. Using results from classical differential geometry, along with careful spectral filtering, we show how topography can be rigorously decomposed into tilings of four shape classes, which provides a natural means of landscape segmentation that highlights channels, basins, domes, and saddles.

We then show through an application to the Oregon Coast Range that calculation of the full curvature tensor reveals details about the geometric evolution of fluvial systems that go beyond that captured by standard slope-area methods. We partition the landscape into segments based on the sign of the curvature invariants, and show how these segments, and the implied shape class distributions, map well on to previously known geomorphic process regimes. Accurately mapping curvature over the entire landscape reveals a remarkable symmetry in Mean curvature that is reflected in both the total landscapes curvature distribution and in the profile curvature measured along ridge/channel networks, which we hypothesize reflect signatures of steady-state fluvial topography.

*Code and data availability.* The code used for data analysis is available at https://github.com/ntklema/TopoCurve_Matlab. The DEM data used in this study is available for download from The National Map at https://apps.nationalmap.gov/downloader/.

*Author contributions.* Conceptualization: NK, LK, Methodology: NK and LK, Visualization: NK and LK, Writing - original draft: NK, LK, and JR.

*Competing interests.* NK is a member of the editorial board of Geomorphica.



*Acknowledgements.* LK acknowledges support from NSF CAREER 1848554. LK acknowledges discussions with Jim Isenberg and with Ian Mynatt, who in different ways inspired interests in the differential geometry of geological surfaces. NK acknowledges that this work benefited from discussions with William Struble and Katharine Cashman.



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
