# Peer review of "Discrete differential geometry of fluvial landscapes"

_EGUsphere, 2025_

## Referee Comment (RC1)

**Review of "Discrete differential geometry of fluvial landscapes"**

Benjamin Kargère

**Overview**

Klema et al. present a new interpretation of topographic "curvature" in the context of coupled fluvial-hillslope landscape evolution. Using the Oregon Coast Range as a case study, they show that, in steep terrain, mean curvature is not equivalent to the Laplacian (despite frequent conflation in the literature) and that mean curvature, together with Gaussian curvature, helps delineate process domains between hillslopes and river networks.

This is an interesting and useful paper, and I think it fits well within the scope of ESurf. I am supportive of it, and the length of this review reflects how much I enjoyed reading it. That said, the foundational claims and their presentation require substantial revision.

**Major Comments**

As stated in the abstract, the major goals of this paper are to introduce mathematical notions of curvature, develop a workflow for measuring the curvature of landscapes, and use this workflow and theory to understand hillslope-fluvial landscapes. Overall, the workflow-related material is clearly presented, including the filtering procedure and figures illustrating the application of curvature-based landscape classification. Given the concerns below, I won't elaborate on these strengths, and I hope this doesn't make the review seem overly negative. I found this paper less than compelling in its presentation of the mathematical framework, its analysis of curvature in the context of geomorphology, and its selection and framing of some of the main findings.

The authors begin by linking curvature to geomorphology through the Laplacian "curvature" (i.e., the divergence of a linear diffusive soil-creep flux) in the stream-power plus linear diffusion model. A major claim of this paper is that the Laplacian is not an accurate representation of curvature in steep terrain. However, the authors fail to note that linear flux laws themselves (and thus the Laplacian) do not accurately represent geomorphic processes in steep terrain. Take, for example, the Andrew-Bucknam/Roering nonlinear flux law:

$$\mathbf{q_s} = \frac{-D\nabla z}{1 - (\frac{|\nabla z|}{S_c})^2}.$$

Performing a Maclaurin expansion in $|\nabla z|$,

$$\mathbf{q_s} \approx -D\nabla z - \frac{D}{S_c^2}|\nabla z|^2\nabla z + ..., \qquad -\nabla \cdot \mathbf{q}_s = D\nabla^2 z + \nabla \cdot \big(D\frac{|\nabla z|^2}{S_c^2}\nabla z\big) + ...,$$

showing that, for shallow slopes, the leading-order term is a linear flux law, the divergence of which returns the Laplacian of z. This linear, local flux law approximation holds for shallow slopes but breaks down at large slopes. This is why hilltop curvature equals the Laplacian and can be used to approximate uplift. As the authors note on line 85, over the horizontal length scales of entire mountain ranges, the Laplacian (linear-diffusion) approximation provides a reasonable estimate of tectonic uplift rates. This holds because the horizontal length scales greatly exceed the vertical length scales, making the shallow-slope approximation appropriate. The shallow-slope approximation is also inherent in the derivation of the stream-power model (Stark and Stark, 2022; Prancevic et al., 2014). I found that the relationship between vertical and horizontal scales and the related shortcomings of the shallow-slope approximation in steep terrain are central to the paper's motivation, yet the authors do not clearly identify these issues.

In Sections 3 and 5 the authors present an extensive review and abstract mathematical derivation of curvature. Although interesting, the discussion of the mathematical history of curvature (Section 3) extends beyond the paper's scope and could be streamlined. Considering the mathematical derivation in Section 5, I would describe the authors' approach as "tell-don't-show," in that the treatment remains abstract with few explicit connections to the landscape evolution equations. For example, explicit mathematical details describing the $(u, v)$ coordinates that follow the surface are not presented. The abstract curvature derivation also contains many inaccuracies. As written, this section adds little practical value beyond the cited references

and, without explicit equations, leaves it unclear how to compute curvature for real landscapes. I have provided several suggestions in the line comments to help streamline the mathematical presentation.

Some of the paper's conclusions are vague and may be misleading. A major example is that the authors label the discrepancy between metrics for the leading-order shallow-slope approximation (Laplacian and drainage area) and metrics for the full three-dimensional (non-shallow) surface (mean curvature and surface area) as "error." It is not "error" so much as that these are not the same quantities. Calling this discrepancy an error also implies that the three-dimensional metrics are more appropriate for this application, yet the authors provide no physical justification for why three-dimensional curvature or surface area are preferable in geomorphology. This is not to deny that the three-dimensional geometric properties are interesting, but their physical utility for landscapes remains unclear. I do not expect the authors to fully resolve this question, but given the lack of evidence, the tone and presentation should reflect this.

A proposed major finding of this paper is an equipartition between positive and negative mean curvature. For this to be "remarkable," as stated in the abstract, there should be some kind of null hypothesis for comparison. However, many statistically random fields (e.g., white noise) would exhibit a similar equipartition. This result would also be unlikely to hold for other ratios of hillslope length to boundary length (the channelization index) (Bonetti et al., 2020; Litwin et al., 2022a; Anand et al., 2023), since, if the domain were zoomed in on a single hillslope, the distribution would change.

The authors interpret the equipartition between positive and negative mean curvatures as evidence that landscapes might minimize surface area. I do not find this suggestion to be persuasive. Minimal surfaces have zero mean curvature everywhere on the surface (pointwise), not in an averaged sense. More generally, optimization or variational principles in physics derive from forces and energies, typically involving the minimization of a potential energy. Minimal surfaces arise due to surface tension, a tangential stress (force per unit length along the interface). Surface tension is absent in landscape evolution, and the relevant forces are gravitational, which are not naturally tangent to the surface but instead point downward. This comment also understates the extensive body of literature on optimization principles in landscape evolution (Rodríguez-Iturbe et al., 1992; Hooshyar et al., 2020; Birnir and Rowlett, 2013; Smith, 2021; Kleidon et al., 2013; Stark and Stark, 2022), which, to my knowledge, makes no mention of surface area. If there were the beginnings of a physical explanation for why landscapes might minimize surface area, I would find this very interesting. In its current form, however, which lacks any physical mechanism, I find the suggestion misleading.

In Section 6.3, the conclusion that surface area may be more correct than drainage area in the sense of the water continuity equation ("potential implications for the interpretation of continuity equations") is misleading. Some of the assumptions in the drainage area formalism include (up to constants of proportionality) that rainfall is constant and is applied via the horizontal $x$-$y$ domain, that horizontal depth-averaged water velocity is constant, that water is routed via the steepest descent of the topography (normal-flow), and that infiltration is neglected (Bonetti et al., 2018; Smith, 2010; Fowler et al., 2007). Given these assumptions, surface area is irrelevant to the continuity equation for (specific) drainage area. In order for surface area to have an effect, one or more of these assumptions needs to be relaxed, which may include Manning velocity parameterization (Gailleton et al., 2024; Smith et al., 1997; Prescott et al., 2025), infiltration/groundwater effects (Litwin et al., 2022b), or possibly orogenic effects.

A final, minor suggestion is to consider noting that drainage area and specific drainage area depend on grid resolution on hillslopes. As a simple example, the drainage area assigned to a topographic maximum equals the grid cell area, and along an idealized planar hillslope it scales linearly with grid spacing. This is not a consequence of filtering, but rather follows from dimensionality. Drainage area has the dimension length squared, whereas the horizontal projection of the contributing region for a point on a hillslope is not necessarily a well-defined area (Kargère et al., 2025; Bernard et al., 2022). Given the widespread use of drainage area to delineate process domains in the literature, it is reasonable to use this metric, but you should note that its value on hillslopes is resolution dependent.

In light of these major recommendations and the length of the manuscript, I suggest streamlining the paper to emphasize what it does best. At its core, this paper presents a substantial amount of interesting and useful material (sufficient for publication), including the workflow, figures, mapping of curvature domains, and the more straightforward connections to between shape classes and geomorphic process domains. The clarification of the distinction between the Laplacian and the mean curvature in steep terrain is particularly useful. Interpreting the more nuanced and exploratory results concerning the physical processes underlying

the shape classes is challenging, and in-depth explanations are clearly outside the scope of the paper. This is not a drawback. It makes the paper more interesting. For the interpretations that are offered, I recommend focusing on those that are well supported by the evidence and by physical reasoning. In this respect, less is more.

**Line Comments**

Before turning to the detailed notes, I want to clarify that many of the following comments (the longer ones) are suggestions and should be taken as such. My goal is not to impose my viewpoint, but offer constructive feedback. For stylistic issues, which occur throughout, I recommend a careful read-through to check punctuation (missing commas), ensure that references to sections, equations, and figures follow the Copernicus style guide, and verify consistent capitalization. I have flagged a few examples, but given the paper's length, it is beyond the scope of this review to note every instance.

5, 9: As noted in the major comments, "systematic error" seems overstated. In the small-slope limit, the Laplacian approximates twice the mean curvature, but this breaks down as slopes increase. Similarly, the difference between upstream surface area and horizontal drainage area reflects distinct quantities, not errors in map-view approaches.

10: The pointwise curvature tensor is a tensor field.

10: Gaussian should be capitalized, "mean" should not be.

12: Complement.

12: As noted in the major comments, I don't think this is an abstract-worthy finding.

17-18: isostasy (not "isostacy").

21: Though it's often described this way in the literature (and I've made this mistake myself) the stream-power model is technically not advection, but a sink term (Bonetti et al., 2020). True advection would take the form $\nabla \cdot (z\mathbf{u})$, where $\mathbf{u}$ is a velocity field.

22-25: "Can be" is used twice.

28: Be more precise. The issue is less about oversimplification and more about overlapping process domains.

33: "Development" is used twice.

76: DEM was already defined.

Eq. (9): The cross product produces the zero vector, which should be typeset in bold.

90: Eq. (1) defines a partial differential equation in $z(x, y, t)$. Therefore, use $\frac{\partial z}{\partial t}$.

132: Per the Copernicus style guide, use Sect., followed by the number in the running text, except when at the beginning of a sentence.

132: As noted in the major comments, the Laplacian of $z$ is proportional to the mean curvature only to leading order when $z$ is $\mathcal{O}(\epsilon)$ relative to the horizontal length scales.

134: "is must"

141: Add a comma after result.

150: Local, not locale.

Figure 2: Use $\Delta$ rather than $d$ to denote a finite, non-infinitesimal change.

163: Citation style

Section 4: This section gets is sandwiched between two sections about curvature, which distracts from the flow. Consider putting it around Sect. 6 or so.

170: Coarser

Figure 2: "dashed lines how." "show"

219: It might be clearer to define the surface in terms of the endpoints of the position vector, rather than defining points on an undefined surface. This definition would be along the lines of: "The surface is the set of endpoints of the position vector, parametrized by the coordinates $(u, v)$, forming a subset of $\mathbb{R}^3$."

220: It would be useful to clarify that each coefficient depends on the $(u, v)$ coordinates, i.e., $r_1(u, v)$, $r_2(u, v)$, $r_3(u, v)$.

224: "the the"

224-225: $u$ and $v$ are scalar coordinates (a chart). The E-W and N-S curves are the level sets where either $u$ or $v$ are constant.

225: Add a comma after "displacement."

225: $ds$ is not a small displacement. Displacement is a vector quantity, whereas $ds$ is a scalar quantity. Refer to the magnitude of the displacement as *infinitesimal* rather than merely "small." By definition $ds = ||d\mathbf{r}||$ is the length of an infinitesimal displacement. Therefore $\mathbf{t} = \frac{d\mathbf{r}}{ds}$ is a **unit** tangent vector, as noted in line 242.

229: Replace the phrase "resultant of" with "resulting from."

230: Add a comma before "with."

232: $ds^2$ is the square of the infinitesimal displacement magnitude (infinitesimal arc length) along the curve.

233: The authors also use $I$ to denote the metric tensor itself (Eq. 33). If this is the chosen convention, be more precise and write $I(d\mathbf{r}, d\mathbf{r})$ to denote the bilinear form $I$ acting on the tangent vectors $d\mathbf{r}$, returning a scalar value.

236: Capitalize "Cartesian."

236-239: The definition of $\alpha$ here distracts from the flow of the explanation.

240: $ds$ is not "any surface curve." It is the infinitesimal increment (line element) along the a curve on the surface.

246: Use 'Eq. (10)' rather than 'equation 10.'

247: As in line 233, refer to the fundamental forms as bilinear operators acting on the infinitesimal displacement vectors, and use parentheses.

255: N-S grid lines, respectively,

**Comments on Section 5**

This section is abstract, and only minimally applied to the problem at hand. I wonder how many in the geomorphology community will have the patience to work through this abstraction. Figure 2 does a good job, but some mathematical 'show-don't tell' could be useful. Below are some mathematical identities that may help connect this derivation to the form of Eq. (1).

A natural place to start might be the $x-y$ 'projected' coordinate system, given that this is the coordinate system for Eq. (1). The (land) surface is defined by the set of endpoints of the position vector

$$\mathbf{r}(x, y) = \big(x, y, z(x, y)\big)^{\mathsf{T}}.$$

This is what you have in Eq. (8), where $x, y, z(x, y)$ are your $r_1, r_2, r_3$. This also appears to be the form of Bergbauer and Pollard (2003). In terms of $(u, v)$:

$$\mathbf{r}(u, v) = \big(x(u, v), y(u, v), z(x(u, v), y(u, v))\big)^{\mathsf{T}}.$$

These coordinate systems are related by a Jacobian:

$$\begin{pmatrix} du \\ dv \end{pmatrix} = \begin{pmatrix} \frac{\partial u}{\partial x} & \frac{\partial u}{\partial y} \\ \frac{\partial v}{\partial x} & \frac{\partial v}{\partial y} \end{pmatrix} \begin{pmatrix} dx \\ dy \end{pmatrix} = \begin{pmatrix} \frac{\partial u}{\partial x} & 0 \\ 0 & \frac{\partial v}{\partial y} \end{pmatrix} \begin{pmatrix} dx \\ dy \end{pmatrix}.$$

This Jacobian matrix is invertible, allowing for changes between the intrinsic and projected coordinates. By Eq. (1), the surface is assumed to be differentiable (no cliffs or overhangs) and therefore $\frac{\partial z}{\partial x}$ and $\frac{\partial z}{\partial y}$ are well-defined. As in Stark and Stark (2022), you may also want to rename the land-surface height to something other than $z$ for clarity. It follows from the assumptions in the paper that:

$$\frac{\partial u}{\partial x} = \sqrt{1 + (\frac{\partial z}{\partial x})^2}, \qquad \frac{\partial v}{\partial y} = \sqrt{1 + (\frac{\partial z}{\partial y})^2}.$$

The metric tensor $(g_{xy})$ in the (x,y) coordinates is:

$$\frac{\partial \mathbf{r}(x, y)}{\partial x} = \big(1, 0, \frac{\partial z(x, y)}{\partial x}\big)^{\mathsf{T}}, \qquad \frac{\partial \mathbf{r}(x, y)}{\partial y} = \big(0, 1, \frac{\partial z(x, y)}{\partial y}\big)^{\mathsf{T}},$$

$$g_{xy} = \begin{pmatrix} 1 + (\frac{\partial z}{\partial x})^2 & \frac{\partial z}{\partial x}\frac{\partial z}{\partial y} \\ \frac{\partial z}{\partial x}\frac{\partial z}{\partial y} & 1 + (\frac{\partial z}{\partial y})^2 \end{pmatrix}.$$

The $g_{xy}$ metric tensor is mapped to $g_{uv}$ (first fundamental form) using the inverse of the Jacobian matrix. If slopes are shallow, then the $g_{xy}$ metric tensor is approximately the identity matrix, since the partial derivatives of $z$ in $x$ and $y$ enter as higher-order terms. To make this explicit, assume $z = \mathcal{O}(\epsilon)$. In other words, the surface height is small compared with the horizontal domain size. Then $\frac{\partial z}{\partial x}$ and $\frac{\partial z}{\partial y}$ are $\mathcal{O}(\epsilon)$, so $F_{xy} = \frac{\partial z}{\partial x}\frac{\partial z}{\partial y}$ are $\mathcal{O}(\epsilon^2) \approx 0$. The deviations of $E_{xy}$ and $G_{xy}$ from 1 are also $\mathcal{O}(\epsilon^2)$. In steep landscapes such as the OCN, the vertical height scale is not negligible compared to the horizontal scale, so the slopes are not negligible, and this expansion breaks down. As a result, the metric tensor here is not approximately equal to the identity matrix.

**Comments on Section 6.1:** The set-up of (Bergbauer and Pollard, 2003) provides an analytical expression for the mean curvature:

$$K_M = \frac{(1 + (\frac{\partial z}{\partial y})^2)\frac{\partial^2 z}{\partial x^2} - 2\frac{\partial z}{\partial x}\frac{\partial z}{\partial y}\frac{\partial^2 z}{\partial x \partial y} + (1 + (\frac{\partial z}{\partial x})^2)\frac{\partial^2 z}{\partial y^2}}{2\,(1 + (\frac{\partial z}{\partial x})^2 + (\frac{\partial z}{\partial y})^2)^{3/2}} = \epsilon\frac{1}{2}\nabla^2 z + \mathcal{O}(\epsilon^3).$$

This form can be compared analytically compared the half Laplacian, which is the leading order term for shallow slope. Given this analytical solution, I wonder whether this section is necessary. I also don't find the inclusion of the D8 algorithm particularly useful. If you prefer to keep these sections, consider moving some of this material to the appendix.

270: The parenthetical note does not aid clarity.

271: Write "Figure" at the beginning of sentences, but Eqn. (19) in the middle.

273: Eqn. (17)

319, 335, 365: Determinant.

383: "the the"

396: "As outlined in Section 3,"

400: It is not clear what is meant by the "$x$" and "$y$" coordinate vectors. I believe you mean the vectors $\partial\mathbf{r}/\partial x$ and $\partial\mathbf{r}/\partial y$, but these have not been defined.

474: No citation.

476: "the the"

564: Citation style.

650: You could mention channel widening (as oin line 610) as well, which to me seems just as important as step-pool morphology, if not more (Bernard et al., 2022; Gailleton et al., 2024).

503, 666: In the current form, the proposed partitioning scheme isn't so much physically justified as geometrically justified (486).

675-679: Given my major comment, it seems unfounded to classify the discrepancy between upstream drainage area and upstream surface area as "error."

**References**

Anand, S. K., Bertagni, M. B., Drivas, T. D., and Porporato, A.: Self-similarity and vanishing diffusion in fluvial landscapes, Proceedings of the National Academy of Sciences, 120, e2302401 120, doi: 10.1073/pnas.2302401120, doi: 10.1073/pnas.2302401120, 2023.

Bergbauer, S. and Pollard, D. D.: How to calculate normal curvatures of sampled geological surfaces, Journal of Structural Geology, 25, 277–289, doi: https://doi.org/10.1016/S0191-8141(02)00019-6, 2003.

Bernard, T. G., Davy, P., and Lague, D.: Hydro-Geomorphic Metrics for High Resolution Fluvial Landscape Analysis, Journal of Geophysical Research: Earth Surface, 127, e2021JF006 535, doi: https://doi.org/10.1029/2021JF006535, e2021JF006535 2021JF006535, 2022.

Birnir, B. and Rowlett, J.: Mathematical Models for Erosion and the Optimal Transportation of Sediment, International Journal of Nonlinear Sciences and Numerical Simulation, 14, 323–337, doi: doi:10.1515/ijnsns-2013-0048, 2013.

Bonetti, S., Bragg, A. D., and Porporato, A.: On the theory of drainage area for regular and non-regular points, Proceedings of the Royal Society A: Mathematical, Physical and Engineering Sciences, 474, 20170 693, doi: doi:10.1098/rspa.2017.0693, 2018.

Bonetti, S., Hooshyar, M., Camporeale, C., and Porporato, A.: Channelization cascade in landscape evolution, Proceedings of the National Academy of Sciences, 117, 1375–1382, doi: doi:10.1073/pnas.1911817117, 2020.

Fowler, A. C., Kopteva, N., and Oakley, C.: The Formation of River Channels, SIAM Journal on Applied Mathematics, 67, 1016–1040, URL http://www.jstor.org/stable/40233429, 2007.

Gailleton, B., Steer, P., Davy, P., Schwanghart, W., and Bernard, T.: GraphFlood 1.0: an efficient algorithm to approximate 2D hydrodynamics for landscape evolution models, Earth Surface Dynamics, 12, 1295–1313, doi: 10.5194/esurf-12-1295-2024, 2024.

Hooshyar, M., Anand, S., and Porporato, A.: Variational analysis of landscape elevation and drainage networks, Proceedings of the Royal Society A: Mathematical, Physical and Engineering Sciences, 476, 20190 775, doi: 10.1098/rspa.2019.0775, 2020.

Kargère, B., Constantine, J., Hales, T., Grieve, S., and Johnson, S.: A fractal framework for channel–hillslope coupling, Earth Surface Dynamics, 13, 403–415, doi: 10.5194/esurf-13-403-2025, 2025.

Kleidon, A., Zehe, E., Ehret, U., and Scherer, U.: Thermodynamics, maximum power, and the dynamics of preferential river flow structures at the continental scale, Hydrology and Earth System Sciences, 17, 225–251, doi: 10.5194/hess-17-225-2013, 2013.

Litwin, D., Tucker, G. E., Barnhart, K. R., and Harman, C.: Reply to Comment by Anand et al. on "Groundwater Affects the Geomorphic and Hydrologic Properties of Coevolved Landscapes", Journal of Geophysical Research: Earth Surface, 127, 2022a.

Litwin, D. G., Tucker, G. E., Barnhart, K. R., and Harman, C. J.: Groundwater Affects the Geomorphic and Hydrologic Properties of Coevolved Landscapes, Journal of Geophysical Research: Earth Surface, 127, e2021JF006 239, doi: https://doi.org/10.1029/2021JF006239, 2022b.

Prancevic, J. P., Lamb, M. P., and Fuller, B. M.: Incipient sediment motion across the river to debris-flow transition, Geology, 42, 191–194, doi: 10.1130/G34927.1, 2014.

Prescott, A. B., Pelletier, J. D., Chataut, S., and Ananthanarayan, S.: An evaluation of flow-routing algorithms for calculating contributing area on regular grids, Earth Surface Dynamics, 13, 239–256, doi: 10.5194/esurf-13-239-2025, 2025.

Rodríguez-Iturbe, I., Rinaldo, A., Rigon, R., Bras, R. L., Marani, A., and Ijjász-Vásquez, E.: Energy dissipation, runoff production, and the three-dimensional structure of river basins, Water Resources Research, 28, 1095–1103, doi: https://doi.org/10.1029/91WR03034, 1992.

Smith, T.: A theory for the emergence of channelized drainage, Journal of Geophysical Research, 115, doi: 10.1029/2008JF001114, 2010.

Smith, T. R.: A Variational Principle for the Integrated Channels and Slopes of Stable Equilibrium Landscapes, Journal of Geophysical Research: Earth Surface, 126, e2020JF006 014, doi: https://doi.org/10.1029/2020JF006014, e2020JF006014 2020JF006014, 2021.

Smith, T. R., Birnir, B., and Merchant, G. E.: Towards an elementary theory of drainage basin evolution: I. The theoretical basis, Computers Geosciences, 23, 811–822, doi: https://doi.org/10.1016/S0098-3004(97)00068-X, 1997.

Stark, C. P. and Stark, G. J.: The direction of landscape erosion, Earth Surface Dynamics, 10, 383–419, doi: 10.5194/esurf-10-383-2022, 2022.